# MR-link-2: pleiotropy robust *cis* Mendelian randomization validated in three independent reference datasets of causality

Adriaan van der Graaf [1,2], Robert Warmerdam [3,4], Chiara Auwerx [1,2,5], eQTLGen Consortium*, Urmo Võsa[6], Maria Carolina Borges[7,8], Lude Franke [3,4] & Zoltán Kutalik [1,2,9] ✉

Mendelian randomization (MR) identifies causal relationships from observational data but has increased Type 1 error rates (T1E) when genetic instruments are limited to a single associated region, a typical scenario for molecular exposures. We developed MR-link-2, which leverages summary statistics and linkage disequilibrium (LD) to estimate causal effects and pleiotropy in a single region. We compare MR-link-2 to other *cis* MR methods: i) In simulations, MR-link-2 has calibrated T1E and high power. ii) We reidentify metabolic reactions from three metabolic pathway references using four independent metabolite quantitative trait locus studies. MR-link-2 often (76%) outperforms other methods in area under the receiver operator characteristic curve (AUC) (up to 0.80). iii) For canonical causal relationships between complex traits, MR-link-2 has lower per-locus T1E (0.096 vs. min. 0.142, at 5% level), identifying all but one of the true causal links, reducing cross-locus causal effect heterogeneity to almost half. iv) Testing causal direction between blood cell compositions and marker gene expression shows MR-link-2 has superior AUC (0.82 vs. 0.68). Finally, analyzing causality between metabolites not directly connected by canonical reactions, only MR-link-2 identifies the causal relationship between pyruvate and citrate ($\hat{\alpha}$ = 0.11, P = 7.2·10$^{-7}$), a key citric acid cycle reaction. Overall, MR-link-2 identifies pleiotropy-robust causality from summary statistics in single associated regions, making it well suited for applications to molecular phenotypes.

The identification of causal relationships in humans is historically done using randomized control trials (RCTs). However, these trials require the separation of individuals into treatment and control groups, which is a burden on the subjects and can be expensive. Additionally, some trials cannot be carried out due to ethical considerations or are simply impossible to perform as there is no suitable way to perturb the trait under investigation.

Observational causal inference attempts to identify causal relationships from observational data by identifying randomization events that occurred naturally in a group of individuals. If performed

[1]Department of Computational Biology, University of Lausanne, Lausanne, Switzerland. [2]Swiss Institute of Bioinformatics, Lausanne, Switzerland. [3]Department of Genetics, University of Groningen, Groningen, The Netherlands. [4]Oncode institute, Utrecht, The Netherlands. [5]Center for Integrative Genomics, University of Lausanne, Lausanne, Switzerland. [6]Estonian Genome Centre, Institute of Genomics, University of Tartu, Tartu, Estonia. [7]MRC Integrative Epidemiology Unit, University of Bristol, Bristol, UK. [8]Population Health Sciences, Bristol Medical School, University of Bristol, Bristol, UK. [9]University Center for Primary Care and Public Health, Lausanne, Switzerland. *A list of authors and their affiliations appears at the end of the paper. ✉e-mail: Zoltan.Kutalik@unil.ch

correctly, observational causal inference can be useful in improving our understanding of the causal relationships that underlie human biology, aiding in the development or repurposing of drugs and treatments.

One observational causal inference technique that is popular in the genetics community is Mendelian randomization (MR)[1,2]. For instance, MR has shown to be effective in identifying the causal relationship between low density lipoprotein cholesterol (LDL-C) levels and coronary heart disease and between alcohol consumption and cardiovascular disease[3,4], while careful application of MR has shown that high density lipoprotein cholesterol (HDL-C) levels are not causally linked to myocardial infarction, evidence that was also corroborated by RCTs[5,6].

A valid MR analysis is done based on three statistical assumptions: i) The relevance assumption, ii) the independence assumption and iii) the exclusion restriction (also known as pleiotropy) (Fig. 1a), described in more detail in (Supplementary Note). Considering the assumptions underlying MR, it is particularly difficult to ensure that the genetic variants that are used as instrumental variables (IVs) are free from horizontal pleiotropy. As it is usually impossible to ensure that a genetic variant only acts through the chosen exposure[7].

MR methods are constantly being developed to ensure MR estimates are robust[8–13]. Generally, these methods use tens to hundreds of independent locations on the genome in a meta-analysis to mitigate violations of the relevance assumption and exclusion restriction with the hope that independent instruments would lead to independent biases which cancel each other out. However, when considering molecular traits, such as RNA and protein expression or metabolite concentrations, there is generally only a single or a handful of associated regions, reducing the robustness of these 'meta-analyzing' MR methods. Improving MR methods for application on molecular exposures is of high priority as these exposures are important causes for disease and are potential drug-targets.

Currently, identification of molecular traits as causes to disease often relies on "closest gene analysis", tools like MAGMA and PASCAL or colocalization methods[14–18]. These approaches have shown that they can identify the correct molecule, but they do not strictly test for causality between two traits. For instance, a closest gene analysis will not identify the context in which a gene has its effect while colocalization analysis will only indicate if two traits share the same causal genetic variant(s). It is not answering the question if one trait is causal to the other or if there is a shared causal confounder. Furthermore, if a molecule under investigation has more than one associated genetic locus, it is difficult to combine the information into a single estimate of relevance. In contrast, MR has the benefit that it identifies a causal relationship, possibly from multiple loci.

Unfortunately, except for some obvious examples, we lack good sources of truth for true causal links and false causal links between (molecular) traits in humans[9,19]. This limits our ability to compare different MR methods, as methodologists usually use simulations and single examples to highlight the strength of their causal inference method.

This study has two goals: i) to introduce a novel summary statistics *cis* MR method that is robust to horizontal pleiotropy and ii) to develop (gold)-standard reference datasets for the validation of *(cis)* MR methods. First, we introduce a summary statistics MR method that is robust to pleiotropy even when only a single region is available for analysis, making it suitable for the analysis of molecular traits as risk factors. We coin the method "MR-link-2" (Fig. 1b). Conceptually, MR-link-2 uses the region surrounding the genetic variant around the IV to estimate the effects of pleiotropy[20]. In contrast to the original MR-link (v1), MR-link-2 does not require individual level data but it is designed to be applied to association summary statistics of the exposure and the outcome, allowing for more widespread applications[20]. The second main novelty lies in the development of benchmarking datasets.

We benchmark MR-link-2 against other *cis* MR methods, two colocalization methods as well as meta-analyzing MR methods (when appropriate) using extensive simulations (Fig. 1c) (Table 1) and three real-world datasets of true and false causal links. In the first real data validation, we create a metabolite network using three sources of curated databases of human metabolite pathways and we assess discriminative performance of each method using metabolite quantitative trait loci (mQTLs) that are derived from four different studies (Fig. 1d). Second, we assess the performance of MR methods on known causal relationships between complex traits, as well as relationships that are unlikely to be causal (Fig. 1e). Third, using new data from the full *trans* mapping of gene expression by the eQTLGen Consortium, we test for the causal relationship between blood cell composition and whole-blood expression levels of cell-type specific marker genes (Fig. 1f)[21].

In all validation datasets, MR-link-2 compares favorably to other methods, exhibiting lower type 1 error (T1E) and often good discriminative performance between true and false causal links, which we attribute to the method's robustness to the presence of horizontal pleiotropy. The pleiotropy robust design of MR-link-2 leads to higher nominal type 2 error rates, as a limited number of true causal links may go undetected. However, when considering results identified outside of the validation datasets, MR-link-2 uniquely identifies regulation between metabolites including an exclusive link that is found in the citric acid cycle highlighting that MR-link-2 can pick up signals missed by other methods.

## Results

### The MR-link-2 method

MR-link-2 is a likelihood function that estimates three parameters based on the exposure and the outcome summary statistics in a region, combined with a reference linkage disequilibrium (LD) matrix (Fig. 1b) (**Methods**). MR-link-2 is now more widely applicable as it does not require individual level data, which was a limitation of the original version[20]. MR-link-2 tests for two parameters using a likelihood ratio test: the causal effect estimate $\hat{\alpha}$, which is of central interest, and the remaining horizontal pleiotropic variance, $\hat{h}_\gamma^2$, which would otherwise violate the exclusion restriction (**Methods**) (Supplementary Note). The modeling of the pleiotropic variance allows MR-link-2 to be robust to violations of LD-induced pleiotropy which can bias MR estimates particularly when using only a single genetically associated region. MR-link-2 replaces the exclusion restriction assumption with two other milder and biologically more plausible assumptions. i) MR-link-2 assumes that all genetic variants in a locus have non-zero genetic effects on the exposure and their mean is zero (infinitesimal model) and all variants (can) have an independent pleiotropic effects on the outcome (InSIDE assumption). ii) MR-link-2 assumes that the LD matrix is measured without error. We therefore estimate that the likelihood function of MR-link-2 can be sensitive to three parameters: i) when the LD matrix is measured with imprecision (either due to small sample size of the reference panel or population mismatch), ii) when the pleiotropic effects are correlated and iii) when there is only a small number SNPs have non-zero causal effect on the exposure and the outcome trait.

### Simulations

To understand the statistical behavior of MR-link-2, we performed simulations of causality in a single genetic region. Our aim was to understand how MR-link-2 performs when the parameters in the simulations are varied, including when assumptions underlying MR (Fig. 1a) and MR-link-2 are violated. We also compare the performance of MR-link-2 with four other *cis* MR methods to understand how each method performs as a function of the simulation parameters. As simulations only approximate the real world, we also apply these *cis* methods to real biological data and compare the

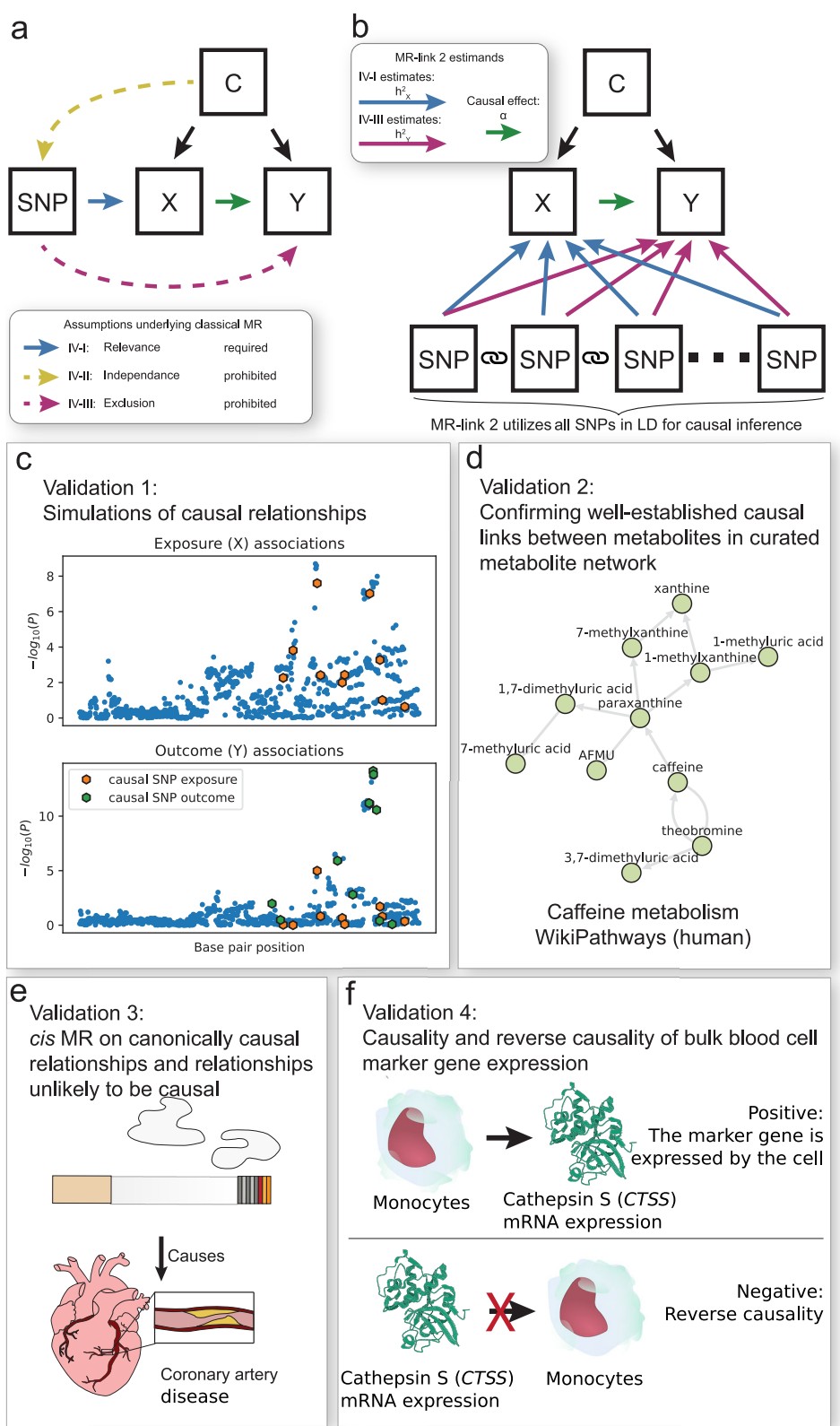

results to a large body of assembled ground truth (**Methods**) (Table 1).

Across 3240 simulation parameter settings, we simulated 1000 instances of exposure and outcome pairs that are genetically regulated by a single region based on LD derived from the UK10K cohort[22]. We varied 6 parameters: the simulated causal effect ($\alpha$), the *cis* heritability of the exposure ($h^2_{\mathbf{X}}$) and the extent of pleiotropy of

the outcome ($h^2_{\mathbf{Y}}$), the number of causal variants ($m_{causal}$) that have a genetic effect on the exposure, and the same number of causal ($m_{causal}$) genetic variant that affect the outcome, their minimum correlation between the causal markers for the exposure and those with direct causal effect on the outcome (min ($r_{causal}$) and imprecision in the LD reference (parameterized by the reference panel size, $n_{ref}$) (**Methods**). Note that in case of sparse genetic effects,

**Fig. 1 | Overview of this study: the assumptions underlying Mendelian randomization (MR), a graphical representation of MR-link-2 method and the four ways we benchmark and compare MR-link-2 to other *cis* MR methods.**
**a** Directed acyclic graph to illustrate the assumptions underlying MR. Single nucleotide polymorphisms (SNPs) are used as instruments to estimate the causal effect between an exposure (X) and an outcome (Y) confounded by C. The blue, yellow and purple arrows highlight the assumptions underlying MR. Black arrows are allowed but are not necessary for correct inference. **b** Graphical representation of the MR-link-2 method. In contrast to other MR methods, MR-link-2 models all the SNPs in a genetic region to simultaneously estimate the (local) *cis* heritability of the exposure (IV-I, $h_X^2$, blue arrows), the total pleiotropic effects on the outcome due to violations of the exclusion restriction assumption (IV-III, $h_Y^2$, purple arrows) and the causal effect $\alpha$ (green arrow) that is robust to violations of IV−III. MR-link-2 requires that linkage disequilibrium is measured in between the genetic variants (chain

symbol). **c**–**f** Validations done to compare MR-link-2 to other methods. (c) First validation is done using simulations. Shown here is a simulated genetic region where an exposure is causal to an outcome. The outcome also contains genetic effects independent of the exposure, which would violate the exclusion restriction (two-sided *P* values come from a univariable linear regression) (IV−III). **d** We perform a second comparison of *cis* MR methods using gold standard metabolite reactions present in curated metabolic networks. For illustration, we show here the human caffeine metabolism from WikiPathways. **e** Validation through canonical causal relationships between complex traits. Shown here, for illustration, is the well-known causal relationship between smoking and coronary artery disease. **f** Final validation tests the ability to decide between forward *vs* reverse causal effects. We utilize the genetics of blood cell proportions to predict their causal effect onto well-known blood cell marker genes. Null causal effects are defined as the reverse direction which should not be causal.

---

increasing the min($r_{causal}$) parameter leads to the violation of the InSIDE assumption.

In simulations with 10 ($m_{causal}$) underlying causal exposure and outcome SNPs and an LD matrix measured with full precision, MR-link-2 has calibrated T1E (min: 0.02, median 0.05, max 0.06), across the range of simulated exposure heritabilities including when there is strong violation of the pleiotropy assumption (Fig. 2a) (Supplementary Data 1). When simulating a large causal effect 0.2 (Fig. 2b) (Supplementary Data 1), we find that increasing the exposure genetic variance increased detection power (up to 1.00), whereas increasing pleiotropy reduced detection power (Fig. 2b). T1E rates generally increased when simulating violations in the MR-link-2 assumptions. MR-link-2 has increased T1E rates when we introduce imprecision in the LD reference. Up to 0.42 when the LD reference is measured only in 500 individuals. This is also seen for other LD dependent methods: MR-PCA T1E = 0.816, MR-IVW LD T1E = 0.782, but as expected, not for MR-IVW, T1E = 0.127, which has lower power in exchange (Supplementary Data 1), when causal genetic variants of the two traits are in very strong LD (up to 0.243 when SNPs are in LD $r^2 > 0.1$ to each other) (Supplementary Data 1). However, MR-link-2 is not dependent on the number of causal SNPs that underlie a trait (max T1E rate = 0.05 when simulating 1 causal SNP for the exposure and 1 causal SNP for the outcome) (Supplementary Data 1). When violating all these assumptions together, the T1E rate increased to 0.84 when simulating a single causal SNP combined with an extremely large $h_Y^2$ of 0.03 (Supplementary Data 1). Nonetheless, this is an extreme and unrealistic situation where all other tested *cis* MR methods (MR-IVW T1E = 0.95, MR-IVW-LD T1E = 0.973 and MR-PCA T1E = 0.96) fail and the T1E is still the lowest for MR-link-2[23,24].

A unique feature of MR-link-2 is that it can identify residual genetic variance in the outcome, which would otherwise be modeled as violations of the exclusion restriction. When simulating minute ($h_Y^2 = 10^{-20}$) pleiotropy and following the MR-link2 underlying assumptions, MR-link-2 does not detect pleiotropy (detection rate minimum = 0.00, maximum = 0.01) with deflated test statistics compared to the expected 0.05 (Fig. 2c) (Supplementary Data 1). However, when simulated pleiotropy is increased above $h_Y^2 = 10^{-4}$ in simulations, MR-link-2 correctly estimates the extent of pleiotropy (Fig. 2c).

To ensure that MR-link-2 is adequately powered, we compared MR-link-2 to three other (*cis*) MR methods (MR-IVW, MR-IVW LD and MR-PCA) as well as to two colocalization methods (coloc and coloc SuSiE) using the area under the receiver operator characteristic curve (AUC) metric[14,15,23–25] (Table 1) (Fig. 2d–h) (Supplementary Data 2). In many cases, we find that the AUC of MR-link-2 is higher, especially when simulating pleiotropy ($h_Y^2 > 10^{-4}$)(Fig. 2f). To understand the influence of each parameter setting on the discriminative ability of each method, we performed ordinary least squares regression with all model parameters as predictors and the AUC of a method as the dependent variable (Fig. 2i) (Supplementary Data 3). Here we see that

the AUC generally decreased for each method as pleiotropy is simulated, with the smallest decline observed for MR-link-2, providing further evidence for robustness to pleiotropy of our method (Fig. 2i) (Supplementary Data 3). Furthermore, we see that the imprecision of the reference panel negatively influences only MR-link-2 and coloc SuSiE (Fig. 2i).

## Using metabolite networks as a source of true causal links

Unfortunately not many causal relationships are established for molecular exposures that could be used to reliably benchmark MR methods[19]. Here, we compile reactions between metabolites as true positive molecular causal relationships[26]. We use these causal relationships as a ground truth, to understand when a particular MR methods fail and to subsequently compare these MR methods to each other.

Our ground truth metabolite network is derived from the human metabolic pathway definitions of KEGG, MetaCyc and WikiPathways[27–29] (Fig. 3a) (**Methods**). Orthogonally, we applied MR/colocalization methods to four mQTL studies comprising of 1,291 harmonized metabolite measurements of 1,035 unique metabolites[30–33] (**Methods**) (Fig. 3a). After harmonization with the pathway definitions, we kept 266 measurements across mQTL studies, representing 154 unique metabolites. One hundred ninety-two metabolite measurements have an mQTL at $P \leq 5 \cdot 10^{-8}$, representing 126 separate metabolites which can be used as exposures to compare their causal effects to the "ground truth" (Fig. 3a–c). Across these 154 unique metabolites, our pathway definitions define 287 individual chemical reactions that can be used as true causal links. (Fig. 3a–c) (**Methods**) (Supplementary Data 4). Comparing pathway definitions between each other, the concordance of MetaCyc and WikiPathways was the highest, while KEGG was less concordant. Indeed, only 34 out of 284 reactions are present in all three pathway databases. Fifty-five reactions are shared by at least two pathway databases and the remaining 194 reactions are specific to single pathway databases (Fig. 3c).

Multiple measurements of the same metabolites across different mQTL studies allows for a bias analysis across MR methods. From 64 unique metabolites across 140 measurements, we perform pairwise causal inference (Supplementary Data 4). Here, the expectation is that the causal estimate of a metabolite on itself is exactly 1.0 and any deviation from this value is considered as bias (Fig. 4a–d). We find that MR-link-2 has the smallest deviation from the expectation ($\alpha$ = 1.00) and thus the lowest estimation bias (mean $\hat{\alpha}$ = 1.027) (Fig. 4a) compared to MR-IVW (mean $\hat{\alpha}$ = 0.935) (Fig. 4b), MR-IVW LD (mean $\hat{\alpha}$ = 0.927) (Fig. 4c) and MR-PCA (mean $\hat{\alpha}$ = 0.873) (Fig. 4d)[23–25] (Supplementary Data 5) (Table 1).

The chemical reactions present in human metabolism are governed by well-established rules, one of which is the principle of *le Chatelier*, stating that an increase in a substrate will increase the

**Table 1 | Feature and requirement comparison of the cis methods used in this study**

| Method | Meta-analysis possible | Uses all SNPs in a region | Requires an LD reference | Robust to horizontal pleiotropy | Tests for |
|---|---|---|---|---|---|
| MR-link-2 | Yes | Yes | Yes | Yes | Causality between two traits and presence of horizontal pleiotropy in the associated region |
| MR-IVW | Yes | No | No[a] | No | Causality between two traits |
| MR-IVW-LD | Yes | No | Yes | No | Causality between two traits |
| MR-PCA | Yes | Yes | Yes | No | Causality between two traits |
| coloc | No | Yes | No | Yes | If a single causal SNP in a region is shared between two traits |
| coloc-SuSiE | No | Yes | Yes | Yes | If any of the causal genetic variants in a region are shared between two traits |

We compare 6 different methods for the identification of cis genetic regulation. We specify if meta-analysis of multiple regions is possible for each method, if the genetic method uses all genetic variants that are available in a genetic region, if the method requires an LD reference and if the method has been designed to handle horizontal pleiotropy. [a] MR-IVW needs LD information only to perform the pruning step.

product of a reaction (**Methods**) (Supplementary Note). Therefore, we expect that a causal estimate that represents a metabolic reaction should be strictly positive, as the causal effect represents the effect of the increase in a substrate. Indeed, when considering Bonferroni significant MR estimates (238,097 testable exposure, outcome and associated region combinations, $P < 2.1 \times 10^{-7}$), all methods identify more positive effects than negative effects (range: 58%–80%) (Fig. 4e–h), with the highest percentage (80%) for MR-link-2 (Fig. 4e), considerably higher than the second-best performing method, PCA-MR (62%) (Fig. 4h) (Supplementary Data 6) (Supplementary Note).

To compare the *cis* performance of all the methods in our metabolite network we assess each exposure locus independently, i.e., we do not meta-analyze any loci together as in Fig. 3b and Fig. 3c. This allows us to make per locus comparisons of coloc and different MR methods, which would be less transparent when meta-analyzed. True causal links are defined as direct reactions (Fig. 3c) (Fig. 4i). As it can be difficult to prove a negative in (human) biology, we utilize a reaction distance metric to define variable sets of false causal links that are increasingly likely to be null (Fig. 4j) (**Methods**). Compared to naively using all available non-causal combinations as false causal links, this approach reduces bias that may be due to causal relationships that exist between understudied metabolites, while also providing multiple AUC measures across different sets of false causal links. As such, this can be viewed as a sensitivity analysis due to imperfect definitions of a true null link dataset (Fig. 4j) (**Methods**). We ensure that the true null set is not too close to true links, by defining true null edges as those with the shortest path being at least five reaction long, while the strictest definition of null edges is defined as the maximum distance in which the number of false causal links is larger than ten (Fig. 4j) (**Methods**).

If we compare the discriminative performance of the MR methods with coloc methods, we find that in aggregate, coloc methods have lower AUC than any MR method used here (Fig. 4k-p) (Table 1)[14,15]. This results in similar discriminative performance across comparisons (Pearson r = 0.997). The AUC of MR-IVW and MR-IVW LD are also correlated (Pearson r = 0.998). MR-PCA and MR-link-2, use the whole genetic region for their inference, regardless of significance of the other genetic variants in the associated region. This approach is beneficial as MR-PCA or MR-link-2 usually (145 out of all 156 AUC comparisons) provide the highest discriminative performance of all methods tested, with MR-link-2 being usually slightly better than MR-PCA: MR-link-2 has the highest AUC in 33 out of 36 comparisons in KEGG (Fig. 4k), 20 out of 33 in MetaCyc (Fig. 4l) and 20 out of 26 in WikiPathways (Fig. 4m) (Supplementary Data 7) (**Methods**).

We reduce uncertainty caused by differences in pathway definitions, by combining pathway references together (**Methods**) (Fig. 4n–p). MR-link-2 remains the method which most often has the highest AUC when all true causal links and false causal links are combined into a union (33 out of 36) (Fig. 3j) (Fig. 4n), when a true causal link and a false causal link is present in at least 2 datasets (12 out of 13) (Fig. 4j) (Fig. 4o). In the smallest true causal link dataset, the intersection of all true causal links and false causal links, PCA-MR has the highest AUC in 10 out of 10 cases (Fig. 4j) (Fig. 4p) (Supplementary Data 7).

Discriminative ability based on an AUC metric is derived from a segmentation of all *P* values or other test statistics. However, investigators researching causal molecular traits generally only consider Bonferroni significant ($P < 2.1 \times 10^{-7}$) results. Therefore, we determine the precision and recall of the MR methods tested at this Bonferroni significance. We find that MR-link-2 has the highest precision in all the pathway comparisons (competing methods have 54–97% of the relative precision of MR-link-2, across all pathways and all methods), with lower recall (between 50–70 % of the recall for MR-link-2, across all pathways) (Supplementary Fig. 1a) (Supplementary Data 8).

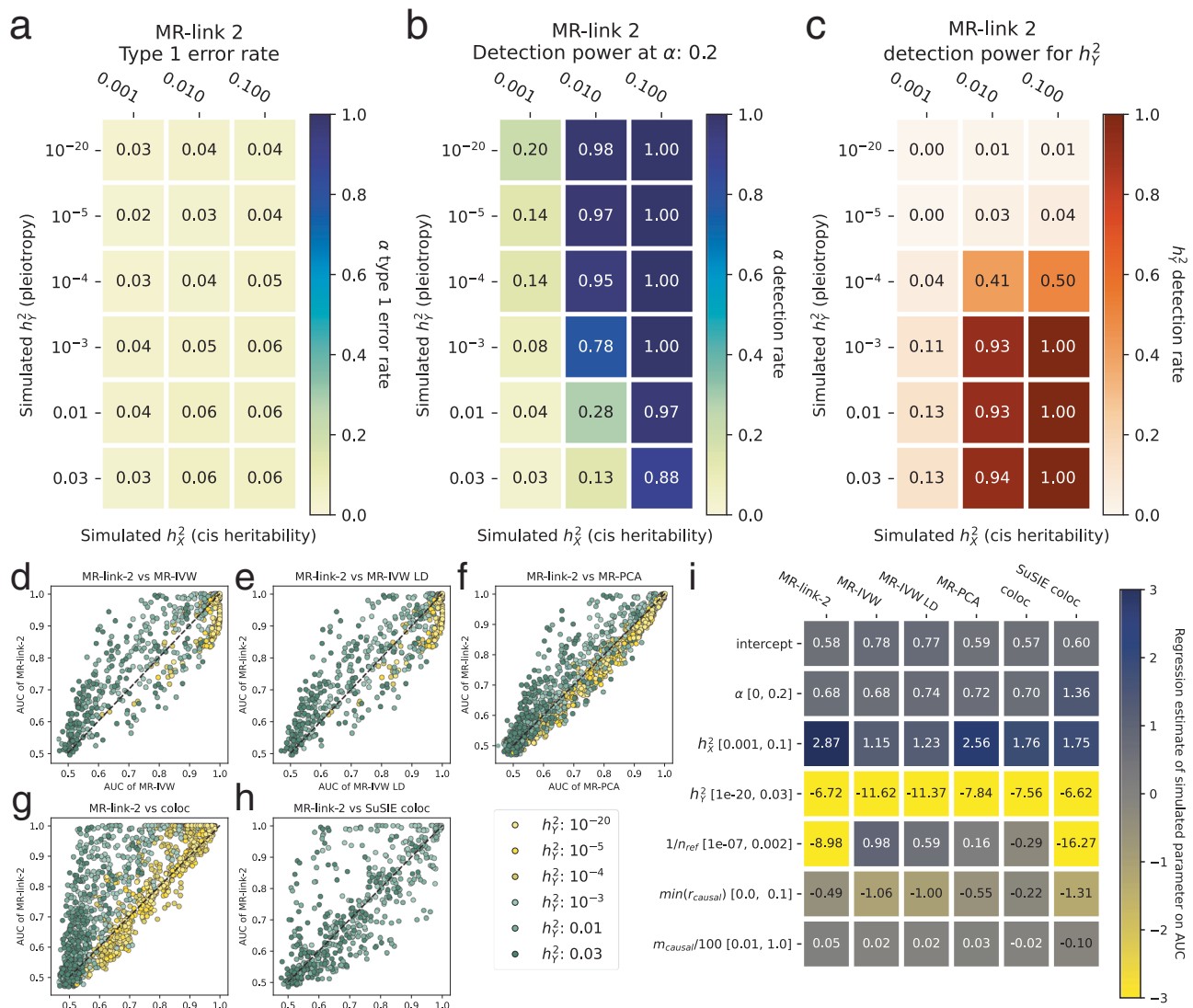

**Fig. 2 | Simulations of MR-link-2 in different scenarios. a** Type I error rate of MR-link-2 in simulations with no causal effect ($\alpha = 0$) and various combinations of exposure genetic variance ($\sigma_{\mathbf{X}}^2$, which is a measure of IV–I) and outcome genetic variance ($h_{\mathbf{Y}}^2$, which violates the IV–III assumption of no pleiotropy). **b** Statistical power in the same simulation scenarios as panel (a) with a simulated **c**ausal effect ($\alpha = 0.2$). **c** The power to detect non-zero pleiotropy by MR-link-2 (testing the pleiotropy parameter $h_{\mathbf{Y}}^2$). The simulation settings are the same as in panel (a), however, here we do not test for a causal effect, rather we test for violations of the IV–III assumptions of no pleiotropy. **d**–**h** The discriminative ability of MR-link-2 and other tested methods between simulations of no causal effect and those with a non-zero causal effect, characterized by the area under the receiver operator characteristic curve (AUC). The AUC values of MR-link-2 are compared to those of other competing methods. Here we also included additional simulation scenarios, where the infinitesimal exposure genetic model is violated (**Methods**). Parameter settings

are only plotted for which both methods successfully estimate at least 750 / 1000 simulation instances in both null and non-null causal effect scenarios. Points are colored by the simulated pleiotropy parameter of $h_{\mathbf{Y}}^2$. The x-axis corresponds to methods as follows: **d** MR-IVW; **e** MR-IVW LD; **f** MR-PCA; **g** coloc; **h** coloc SuSIE. (**Methods**) (Supplementary Data 2) (**i**) A heatmap of (multivariable ordinary least squares) regression coefficients for each method when AUC is regressed on various model parameters. This allows identification of the impact of each simulation parameter on the AUC of each method. The simulated range of each parameter is shown in brackets. $1/n_{ref}$: represents the precision of the linkage disequilibrium reference used in this study, i.e. the inverse of the reference panel size. $min(r_{causal})$ represents the minimum correlation between the causal SNPs and SNPs with direct effect on $Y$. $m_{causal}/100$ represents the number of causal SNPs selected in the region divided by 100 to ensure comparable regression coefficient scales (**Methods**).

Up to now, we analyzed single locus estimates in isolation; however, MR estimates can be meta-analyzed together using the inverse of their variance estimate as weights (not to be confused with MR-IVW methods) (Fig. 3b) (Fig. 3c) (Table 1) (**Methods**).

When meta-analyzing results, discriminative performance of MR-link-2 is less pronounced over the other causal methods tested in this manuscript (116 out of 148 comparisons MR-link-2 does not have superior AUC) (Supplementary Fig. 2) (Supplementary Data 9). Moving away from the true causal links and negative causal links in the pathway definitions and considering the broader Bonferroni significant (49,335

exposure-outcome combinations, $P < 1.0 \times 10^{-6}$) results of these estimates weighted across associated regions, the precision remains highest for MR-link-2 in 4 out of 6 pathway comparisons and more than 99% of the relative precision otherwise. Unlike in the regional estimates, the recall of MR-link-2 is similar compared to competing MR methods: MR-link-2 has a recall that is 72–97% of the highest competing method (Supplementary Fig. 1b) (Supplementary Data 8).

When applying MR, it is usually recommended to apply multiple methods to ensure that conclusions are not based on the assumptions of a single method. Since our focus was to benchmark these methods

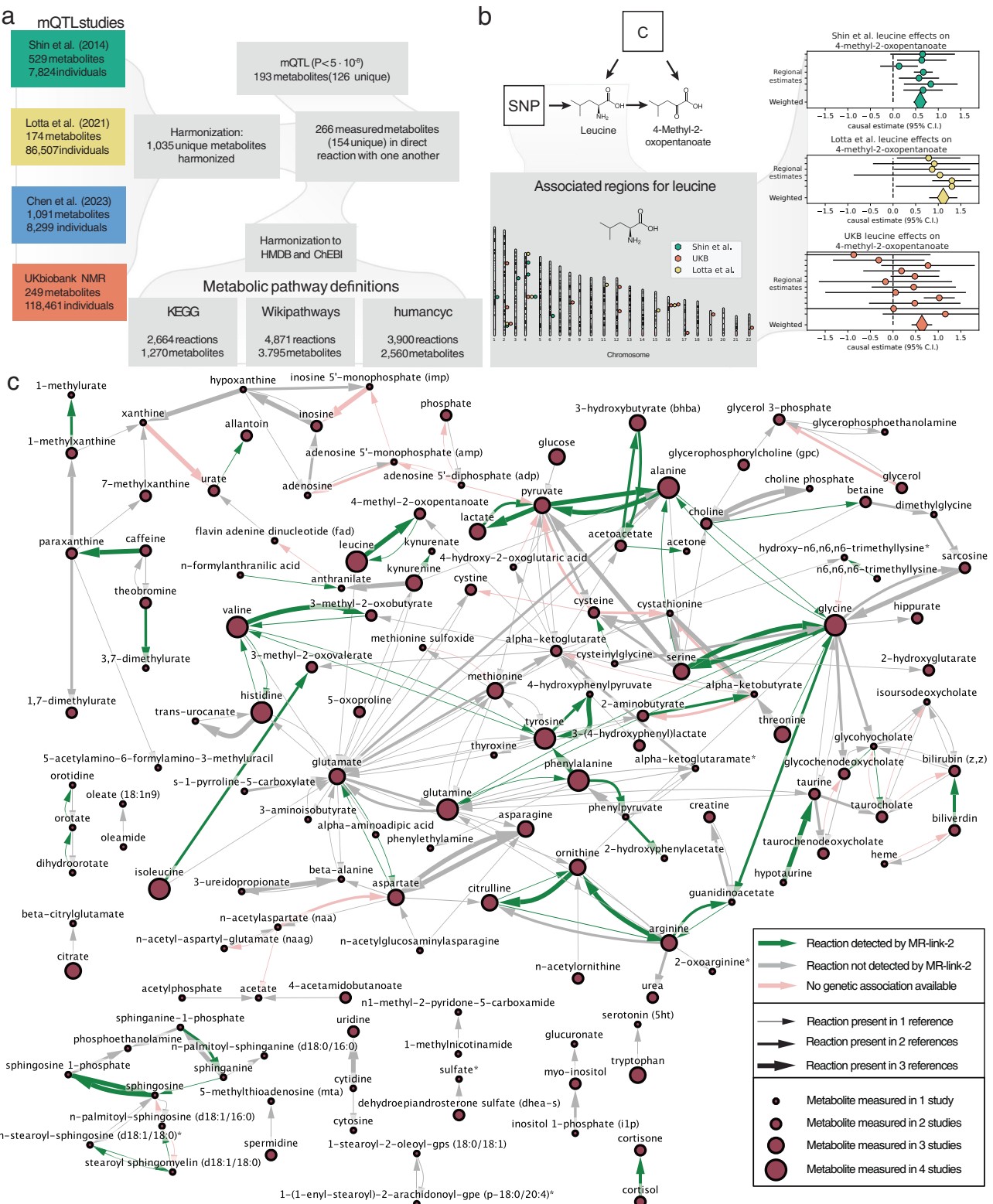

on reference data sets, we were more interested in the global patterns of result overlaps across methods[34]. We compare the 848 Bonferroni significant MR-link-2 results weighted across regions to the Bonferroni significant results of other tested MR methods, we find that MR-link-2 shows less similar results compared to other methods, with the lowest Jaccard index (min = 0.34, max = 0.45, lowest other Jaccard index = 0.66 between MR-PCA and MR-IVW) in a pairwise comparison of all four MR methods tested (Supplementary Fig. 3).

MR-link-2 identifies 188 causal relationships at Bonferroni significance that are not found by other methods. This is more than for any competing method: 19 for MR-IVW, 4 for MR-IVW LD and 95 for MR-PCA. Sixteen of these 188 unique MR-link-2 estimates are found in the pathway databases used in this study whereas this number is only 3 out of 95 for MR-PCA, 2 out of 15 in MR-IVW LD and none of the 15 links unique to MR-IVW (Supplementary Data 10).

**Fig. 3 | Metabolite quantitative trait loci (mQTL) studies used in this analysis, an example MR analysis and the true causal links and true positives identified in this study. a** Chart depicting the metabolites and their mQTLs used in this study. We utilized four mQTL studies whose studied metabolites were harmonized into 1035 consensus metabolites. To create ground truth causal links between these metabolites, we used three pathway definitions. Overlapping mQTL studies with the metabolite databases resulted in 266 metabolite measurements across studies. Metabolites can be measured in multiple studies, leading to 154 unique measured metabolites. In Mendelian randomization (MR), an exposure (a substrate in a reaction) needs to have at least one mQTL available, resulting in 193 (126 unique) metabolites with at least one SNP ($P \leq 5 \cdot 10^{-8}$). This is not a requirement when the metabolite is the outcome. **b** Example MR result for the reaction between leucine and 4-methyl-2-oxopentanoate (supported by three databases). Leucine has genetic associations in 3 out of 4 mQTL studies where it was measured. We use SNPs in the associated regions for leucine as instruments to estimate the causal effect of leucine on 4-methyl-2-oxopentanoate. For brevity, causal estimates are only shown when the outcome is measured in Shin et al. All regional causal estimates (round circles) can be meta-analyzed into a weighted estimate (large diamond) for a joint causal estimate. **c** The ground truth positive causal relationships between metabolites extracted from 3 databases, containing 287 reactions across 154 metabolites. Causal estimates outside the pathway definitions are not shown. The size of the nodes represents the number of measurements. Arrow width represents the occurrence of the reaction in the metabolic pathway definitions. The color denotes if a reaction was found or not. Green: The reaction was Bonferroni significant ($P < 1.0 \times 10^{-6}$) for MR-link-2 in at least one study combination when meta analyzing the estimates across the reaction (the weighted estimate from panel **b**). Grey: The reaction was not Bonferroni significant for MR-link-2. Pink: The substrate in the reaction does not have associated regions, meaning that there is no data for causal estimation.

## Biological interpretation of causal relationships between metabolites

We further explored the 188 Bonferroni significant MR-link-2 estimates (weighted across regions) between metabolites that are not reported by any of the three pathway references. (Supplementary Data 10). Even though they may not necessarily be direct chemical reactions, causal relationships that MR-link-2 uniquely identified can be integrated into human metabolism. For instance, MR-link-2 is the only MR method tested that identifies a bidirectional causal relationship between pyruvate and citrate ($\hat{\alpha} = 0.11$, P = $7.2 \cdot 10^{-7}$. reverse = $\hat{\alpha}$: 0.08, P = $3.3 \cdot 10^{-9}$). These are the only Bonferroni significant metabolite pairs which are found in the citric acid cycle pathway (HMDB)[35,36]. Another striking example in energy metabolism is the negative bidirectional causal relationship identified between lactate and acetoacetate ($\hat{\alpha} = -0.25$, P = $5.85 \cdot 10^{-10}$. reverse: $\hat{\alpha} = -0.34$, P = $1 \cdot 10^{-8}$). MR-link-2 is the only method that identified the negative causal relationship between the anaerobic fermentation pathway represented by lactate concentrations and the aerobic respiration pathway, represented by acetoacetate concentrations[37,38] (Supplementary Data 10). MR-link-2 is also the only method that identifies the positive causal relationship between two unsaturated fatty acids, (1-(1-enyl-stearoyl)-2-arachidonoyl-gpe (p-18:0/20:4) on cholesterol: $\hat{\alpha} = 0.30$, P: $4.3 \cdot 10^{-7}$ and n-stearoyl-sphingosine (d18:1/18:0) on cholesterol $\hat{\alpha} = 0.33$, P = $1.8 \cdot 10^{-8}$). These two causal relationships recapitulate the role cholesterol has in stiffening membranes that contain unsaturated fatty acids[39,40] (Supplementary Data 10).

MR-link-2 further exclusively identifies the causal relationship between 3-hydroxybutyrate (3HB) and lactate ($\hat{\alpha} = 0.21$, P = $4.01 \cdot 10^{-9}$). This relationship has been corroborated by a clinical study that infused 3HB in heart failure and reduced ejection fraction patients[41]. After 3HB infusion, lactate levels were increased. This is hypothesized to be due to competitive inhibition of 3HB that is preferentially used over pyruvate as a substrate for oxidative phosphorylation, the excess pyruvate is then converted into lacate[42].

## Applying *cis* MR methods to complex traits

Using a second independent benchmarking dataset, with the goal to ensure that MR-link-2 is not only effective at identifying causal relationships between molecular traits, we tested the *cis* MR performance in a set of true and null relationships between complex traits. For this we offset complex trait combinations that are 'considered causal' with those that are considered 'implausible or unsupported' and 'considered non-causal' as defined by Morrison et al.[9] (**Methods**).

Since complex traits do not rely on MR performed in a single region, we additionally explored the performance of state-of-the-art non-*cis* MR methods, such as the MR-PRESSO and the MR-APSS methods[7,43], to five trait combinations that are unlikely to be causal, e.g., outcomes that precede the risk factor in time, such as adult LDL-C levels impacting childhood onset asthma (COA) (**Methods**). Only one false positive link is identified at nominal significance (by MR-IVW and MR-IVW LD, diastolic blood pressure on COA, $P = 0.023$ and $P = 0.019$ respectively) (Supplementary Data 11) (**Methods**). All methods falsely identify the 'non-causal' relationship between HDL-C and coronary artery disease (CAD) as well as between HDL-C and stroke, which is notoriously difficult to accurately estimate through univariable MR methods[9] (Fig. 5a). In the positive control analysis, all MR methods including MR-APSS and MR-PRESSO, identify all causal relationships at nominal significance, albeit these latter have larger confidence intervals than *cis* MR methods (Supplementary Data 11).

*Cis* MR methods can provide a per locus causal estimate both on the true positive and true negative combinations. When determining the per locus detection rate at nominal significance ($P < 0.05$) for the *cis* MR methods (excluding MR-APSS and MR-PRESSO as they do not provide regional estimates), MR-link-2 has a consistently lower median T1E rate: MR-link-2: 0.096, MR-IVW : 0.142, MR-IVW LD : 0.160, MR-PCA : 0.163 (Supplementary Data 12) (Fig. 5a). The lower T1E rate could be interpreted as lower power and indeed the detection rate is lower when analyzing causal relationships that are 'considered causal' (MR-link-2 median detection rate per locus = 0.23, MR-IVW = 0.272, MR-IVW LD = 0.266, MR-PCA = 0.255) (Fig. 5b). The median power over median T1E ratio is highest for MR-link-2: 2.4 compared to 1.56 -1.9 for other *cis* methods.

After meta-analysis, regional estimates can deviate from the weighted causal effect. Indicating that the associated region seems to deviate from the meta-analyzed estimate, which is partly due to violations of the "no-pleiotropy" assumption. To test the relevance of MR-link-2's pleiotropy parameter estimate, for each region we compared its value to the absolute deviation between the regional IVW-MR estimate and the meta-analyzed version across all regions. The latter is expected to be larger for regions where the pleiotropy assumption is more violated. We found a strong spearman correlation of 0.21 ($P = 1.5 \times 10^{-235}$) between nominally significant estimates of $\hat{h}_Y$ and the MR-IVW absolute deviation from the weighted estimate (Fig. 5c) (Supplementary Data 13), suggesting that MR-link-2 detects region-specific pleiotropy accurately (without the knowledge of estimates from other regions). This correlation also holds when the deviations are computed for other methods, including MR-link-2 (Supplementary Data 13).

Moreover, when meta-analyzing all loci for all complex trait combinations analyzed in this study, we find that MR-link-2 has substantially lower heterogeneity in terms of Cochran's Q statistic (median for MR-link-2 = 582, lowest competing = 1021) (**Methods**) (Fig. 5d) (Supplementary Data 11).

We further applied all *cis* MR methods in "drug-target-MR" settings. Exposure-outcome MR is performed using instruments from the vicinity of one gene. As positive controls, we used the following three gene-exposure-outcome combinations: LDL-C→CAD instrumented from HMGCR[44], IL18→atopic dermatitis instrumented from IL18[45] and C-reactive protein→COVID-19 instrumented from IL6R[46,47]. As negative

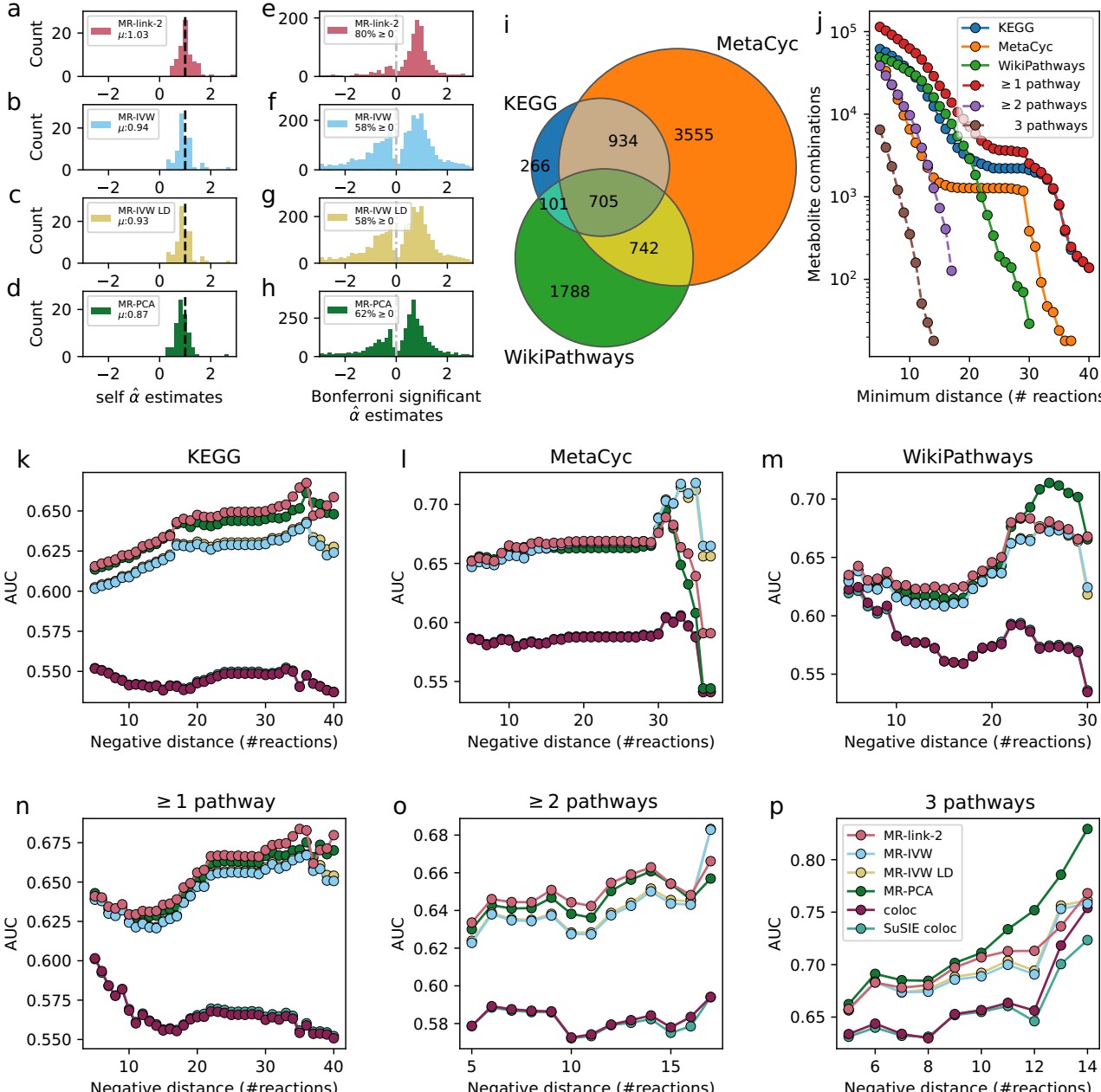

**Fig. 4 | Comparison of different *cis* MR methods through effect size analysis, the true and false causal link datasets used for a comparison of discriminative ability of the metabolites in this study.** Causal effects are estimated for an exposure for each associated exposure region, testing single region results for each region. **a**–**d** The causal effect estimates of the Mendelian randomization (MR) methods tested in this study, when comparing nominally significant ($P \leq 0.05$) estimates between a metabolite on itself using two different mQTL datasets, when they are not included in the true positive dataset. The mean ($\mu$) of a self-estimate is expected to be 1.0. panels represent different methods: **a** MR-link-2 (79 comparisons), **b** MR-IVW (80 comparisons), **c** MR-IVW LD (82 comparisons) and (**d**) MR-PCA (80 comparisons). **e**–**h** Distribution of Bonferroni significant ($P \leq 2.1 \cdot 10^{-7}$) regional causal effect estimates. We report percentage positive effect size estimates, these likely represent direct metabolic reactions, as substrate to product reactions should have positive effect. **e** MR-link-2 (1242 combinations), **f** MR-IVW (3218 combinations), **g** MR-IVW (3373 combinations) and (**h**) MR-PCA (3229

combinations). **i** A Venn diagram representing the number of true causal link combinations used for the regional results in this study per pathway definition. True positives are metabolites (one for each associated exposure region) that are one reaction apart. **j** Negatives used in this study. We define the link between two metabolites as a negative when separated by at least *m* reactions in the full metabolite graphs created from the databases (combinations with more than 10 links). **k**–**p** The area under the receiver operator characteristic curve (AUC) of *cis*-MR and colocalization methods benchmarked against different databases (**k**–**m**) and database combinations (**n**–**p**). Only considering comparisons with more than 10 negatives (same as panel **j**) per positive definition (same as panel **i**). When there is no SuSIE coloc estimate available for a region, the original coloc estimate is used. True causality and false causality: **k** from the KEGG pathway, **l** from the MetaCyc pathway, **m** from the WikiPathways pathway, **n** present in any pathway definition, **o** present in at least two pathway definitions, **p** shared in all pathways.

controls, we used C-reactive protein →Type-2-diabetes 19 instrumented from IL6R[48], LDL-C→BMI instrumented from PCSK9[49] and C-reactive protein →coronary artery disease instrumented from CRP[50], based on established and disproven causality from these loci. After

Bonferroni correction, only MR-link-2 ($P = 1.2 \times 10^{-9}$) and MR-PCA ($P = 1.7 \times 10^{-7}$) identify the causal link between LDL-C and CAD instrumented from HMGCR after multiple testing correction across 104,007 pairwise regions of these complex traits ($P < 4.8 \times 10^{-7}$)

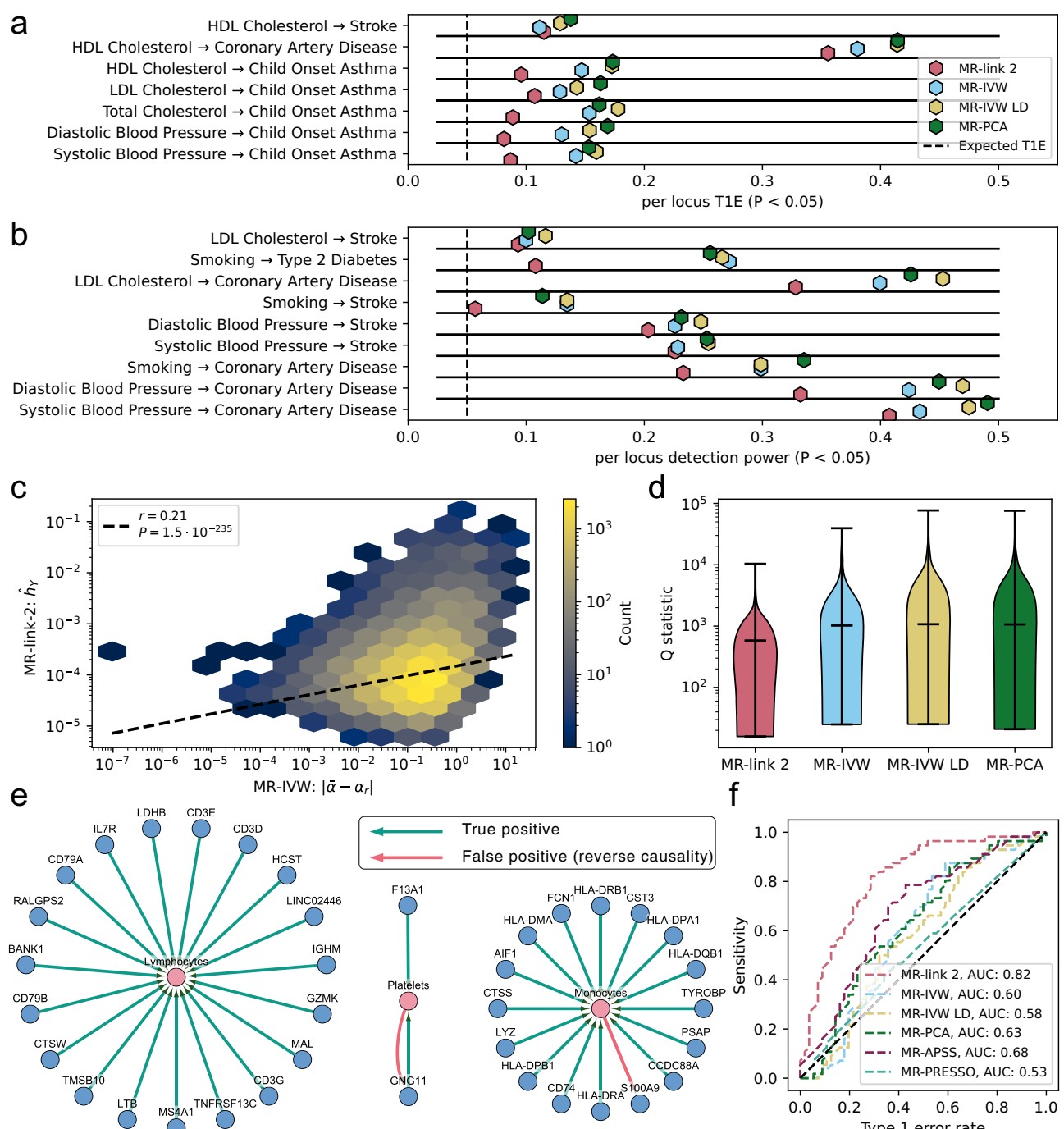

**Fig. 5 | Analysis of different MR methods on canonical causality and the causality between blood cell traits. a** The per locus detection rate (at $P < 0.05$) for phenotype combinations that are not considered causal or are unlikely to be considered causal by Morrison et al. **b** The per locus detection rate (at $P > 0.05$) for phenotypes that are considered causal. **c** Pleiotropic estimates ($\hat{h}_Y$, where $\hat{h}_Y$ $P < 0.05$) (y-axis) compared to MR-IVW absolute regional ($\alpha_r$) estimate deviation from the meta-analyzed one ($\bar{\alpha}$) (x-axis). The $r$ correlation coefficient is the Spearman correlation. The regression line is from linear regression n $\log_{10}$ transformed comparisons. Due to a large number of points in the plot, the points are shown as a density. **d** Violin plot of the heterogeneity statistics of 306 complex trait to complex trait comparisons for the MR methods tested in this study. Upon meta-analysis of all

the pairwise phenotype combinations in (**a**) and (**b**), we plot the Q statistic for each method ($\log_{10}$ scale). The bars and whiskers in the plot refer to the minimum, median and maximum heterogeneity value. **e** Blood cell type and eQTL analysis results. MR-link-2 Bonferroni significant ($P < 5.15 \cdot 10^{-4}$) causal links between cell type concentrations and the RNA expression of their respective marker genes (Supplementary Data 18). Green colored arrows indicate the cell type influences the RNA gene expression in blood causally. These are considered true causal links. The red arrows indicate an (incorrect) causal link between the gene expression and the blood cell type marker, indicating reverse causality. **f** Area under the receiver operator characteristic curve for the cell type directionality analysis for all MR methods tested in this study based on the reported $P$ value of the method.

(Supplementary Data 14). When using a more lenient $P$ value threshold of only these 6 positive and negative controls ($P < 0.05 / 6$), all 3 positive controls are identified by MR-IVW LD and MR-PCA and 2 / 3 positive controls are detected by MR-link-2 and MR-IVW. This

undetected positive control (C-reactive protein→COVID-19 instrumented from IL6R) has little evidence of causal variant colocalization (coloc PP4: $1.0 \times 10^{-16}$) in the locus. This finding may seemingly contrast with the clinically validated efficacy of tocilizumab in severe COVID-19,

which targets the IL-6 pathway. However, it is possible that CRP levels do not fully capture the relevant immunopathological mechanisms induced by IL-6 receptor blockage in COVID-19. Only at the lowest thresholds of significance (0.05/6) was this combination significant in MR-IVW LD and MR-PCA.

## Using reverse causality as true causal links from eQTL information

As a final independent true positive dataset, and to ensure that MR-link-2 does not identify reverse causality excessively, we applied the studied *cis* MR methods and MR-APSS and MR-PRESSO to test how often we can identify the correct causal direction between whole blood bulk gene expression levels and blood cell composition phenotypes[51–53] (Fig. 1f). Our assumption is that when analyzing a mixture of cells such as blood, the differences in the concentrations of certain cell type causally affect the gene expression of respective marker genes (i.e., the observed expression quantitative trait locus (eQTL) effect is caused by genetically regulated cell type differences), whereas the opposite direction is a false positive. These marker genes are derived from reference single cell RNA expression experiments that are typically used to identify cell types from untargeted assays[54]. We test the bidirectional causal relationship between RNA expression of 73 cell-type marker genes and 12 types of peripheral blood mononuclear cells (PBMCs) measured in up to 563,085 individuals of European ancestry[54]. Using newly generated RNA expression eQTLs from the eQTLGen consortium that contains a genome-wide *cis* and *trans* eQTL summary statistics from 19 cohorts and 14,855 individuals[55] (**Methods**) (Supplementary Note).

Upon the meta-analysis of MR estimates across associated regions for 42 true (cell type to gene expression) and 54 false (gene expression to cell type) causal combinations, we find 38 Bonferroni significant ($P < 1.9 \times 10^{-4}$) MR-link-2 comparisons. All estimates have a positive causal effect direction estimate (Supplementary Data 15). Indeed, we find that 94.7% (all but two) of the Bonferroni significant MR-link-2 estimates are in the correct direction (from cell type to RNA expression) (Fig. 5e). Interestingly, in these Bonferroni significant estimates, one false positive is bidirectional, meaning that MR-link-2 identifies both the true causal link and the false causal link (Fig. 5e). In comparison with the other tested *cis* MR methods, we find that MR-link-2 has higher discriminative ability (MR-link-2 AUC: 0.82, best competing: 0.68) to identify the correct effect direction based on the MR methods causal estimate P value (Fig. 5f) (**Methods**).

We were intrigued by the reverse causal direction that MR-link-2 identified between *S100A9* expression and monocyte concentrations. *S100A9* is considered a marker gene for monocyte concentrations based on single cell experiments[54]. MR-link-2 only identifies the reverse effect: *S100A9* increases monocyte concentration with a larger effect (*S100A9* as exposure: 1 locus, $\hat{\alpha}$ = 0.112, P = $3.4 \cdot 10^{-6}$, monocyte count as exposure 514 loci, $\hat{\alpha}$ = 0.06, P = $2.8 \cdot 10^{-3}$) (Fig. 5f) (Supplementary Data 15). Based on a literature review, the causal effect direction of *S100A9* may actually have been correctly estimated by MR-link-2, as *S100A9* has been shown to promote accumulation of leukocytes from mouse knockout experiments[56] as well as inhibiting dendritic cell differentiation[57]. Dendritic cells are a class of monocytes, which upon differentiation, migrate to non-blood tissues, reducing monocyte blood concentrations. These results suggest that *S100A9* expression promotes monocyte accumulation in whole blood and might represent a true causal link, as found by MR-link-2.

## Discussion

In this study, we present a new *cis* MR method: MR-link-2, which can perform pleiotropy robust MR in a single region when summary statistics and an LD reference are available. To our knowledge, MR-link-2 is the only summary statistics *cis* MR method that explicitly models

pleiotropy. MR-link-2 provides estimates of causal relationships through modeling the effect of all SNPs in an entire *cis* region on an exposure and an outcome. We have tested MR-link-2 in four different validation datasets, three of which being based on real data. When compared to competing *cis*-MR and colocalization methods, MR-link-2 has better discriminative ability and lower T1E. Furthermore, MR-link-2 uniquely identified compelling biological examples, such as the negative causal relationship between lactate and acetoacetate which contrasts anaerobic and aerobic energy pathways in humans, respectively, and the relationship between 3HB and lactate, which is a more cryptic causal relationship that has been validated in human trials. Together, these results illustrate the ability of MR-link-2 to help shape our understanding of the underpinnings of molecular mechanisms in humans, including those underlying disease.

Our simulations suggested that MR-link-2 is sensitive to scenarios with strong LD between causal SNPs, especially when the reference LD matrix is measured with high uncertainty, in which MR-link-2 has high T1E, which is a commonality it shares with all other tested methods. Furthermore, our simulations are somewhat limited as we do not simulate directional or correlated pleiotropy[8,9,11,58]. We found it important to show both simulation and real data applications, as each has its own weaknesses.

In real data validations MR-link-2 exhibits lower T1E rates compared to other methods while retaining good discriminative ability. The application to real-world validation datasets show that MR-link-2 is robust to the presence of horizontal pleiotropy in a locus. We make this conclusion based on the lower T1E in the per locus estimates of complex traits, the closer agreement with *le Chatelliers* principle in metabolites, the correlation of the pleiotropy parameter $\hat{h}_{\mathbf{Y}}^{2}$ with regional deviation from meta-analysis, the lower heterogeneity of estimates in our complex trait application and the identification of causality in the correct direction in the blood cell count and eQTL analysis.

This does not mean that MR-link-2 is robust to all violations of the assumptions underlying MR. In cases of extreme pleiotropy, e.g., the HDL-C to CAD analysis, T1E rates are increased. In less extreme cases, we expect the per locus T1E to be 0.05, while our median per locus T1E is 0.09 in the non-causal complex trait analysis. This analysis indicates that accounting for pleiotropy could be improved by allowing for multiple exposures in the model, which could be a natural extension of MR-link-2. Compared to other methods, MR-link-2 does not always provide superior estimates. For instance, in the *cis* locus metabolite analysis, MR-link-2 has superior AUC over other methods in 105 / 156 (76%) of comparisons. MR-link-2 also missed one (IL6R instrumented CRP levels and COVID-19) out of three true positives in the drug-target-MR analysis at lenient multiple testing thresholds.

Taken together and given that MR-link-2 i) provides meaningfully different results from the other tested methods in this study and ii) has lower T1E than other methods, we believe that MR-link-2 can be useful both as a standalone method as well as for secondary validation. Even though the MR methods tested in this work are developed to be principally for the analysis of *cis* associations, they still perform well when meta-analyzing multiple *cis* estimates together. We have found no limitation to the number of regions that MR-link-2 estimates, enabling its application to complex exposures.

The true and null causal relationships from the reference datasets used in this study can be discriminated by genetically informed causal inference methods. Unfortunately, these reference datasets remain imperfect, which is illustrated by the limited agreement between metabolite pathway references in the metabolite datasets. We purposefully did not include metabolite pairs that are more than 1 reaction away from each other, as such links would represent an order of magnitude smaller causal effects, which are undetectable given the current sample sizes. Hence such seemingly true links would only be detected by mistake by any method. Another striking example of

imperfect reference datasets relates to *S100A9*, for which we hypothesize that altered gene expression is causally changing blood composition. This is at odds with our initial assumption that changes in blood composition are causal to changes in the expression of cell marker genes.

Nonetheless, the benchmarking data developed in this study provide a reasonable ground-truth for testing causal inference methods. In the future, the community should increasingly seek to refine and broaden these datasets to facilitate the development and benchmarking of new causal inference tools. One important step towards this is to build causal datasets in a tissue specific way. The current ground truth datasets will contain some tissue specific links that will be difficult to detect. Indeed, the limited tissue specificity in the metabolite analysis likely contributes to the relatively low recall of all causal edges. Improvement of the tissue specificity in these datasets may contribute to better discriminative ability of each method.

In this study we use summary statistics of European ancestry individuals when they are available. We justify this decision based on sample sizes and availability of a large and representative LD reference. It is important to note that our methodology is not limited to individuals of European ancestry. We even hypothesize that a more diverse association panel would make MR-link-2 more powerful as the diversity of the available LD panel will allow for better distinction between causal effect and horizontal pleiotropy which we believe is paramount to correct causal inference. The availability of more multi-ancestry population studies is likely to improve our understanding of causality, with the caveat that the LD panel should match the ancestry composition from which the summary statistics were derived.

In all the validations performed in this study, MR-link-2 has shown to be a promising tool to study exposures with dominantly *cis*-associations and leads to improved identification of causal links between molecular trait, which will – in turn – facilitate shedding light on the molecular basis of complex traits.

## Methods
### Research ethics compliance
This work includes data that is derived from human participants, all whom provided informed consent for their analysis. All studies have been approved by research ethics committees. We analyzed three categories of human-derived data: i) publicly available summary statistics of human traits, ethics and consent statements of which can be found in the source publications, ii) the UK10K genotype reference which we accessed under data access agreement. Ethical approval and informed consent statements can be found in the UK10K source publication[22] and iii) summary statistics from the eQTLGen Consortium. These research activities have been carried out under the ethical approval nr. 1.1-12/655 and its extension 1.1-12/490 by the Estonian Committee on Bioethics and Human Research (Estonian Ministry of Social Affairs).

### Mendelian randomization and colocalization methods
**Existing cis genetic methods used in this study.** We use six *cis* genetic methods in this study to identify if two phenotypes are causally related to each other. Two methods are colocalization methods and four methods are MR based methods.

We use two versions of colocalization, namely coloc[14] and coloc-SuSIE[15] (R v4.3.2 coloc package version 5.2.3). Both methods test for the following question: Are the causal variants of the traits shared? coloc assumes that there is a single causal variant in the locus, whereas coloc-SuSIE relaxes this assumption by identifying independent SNPs and subsequently performing a conditional coloc analysis per variant. We define detection of a causal variant sharing as having a coloc posterior probability (PP4) larger than 0.9 for the 4th hypothesis (two traits share a causal variant). As coloc-SuSIE may estimate multiple causal effects, we take the maximum PP4 across analyses (Table 1).

We use three existing (*cis*) MR methods and introduce one new MR method: MR-link-2. The three existing MR methods used are MR-IVW[24], MR-PCA[23] and MR-IVW LD[23]. We used the 'mr_ivw' and 'mr_wald_ratio' functions (for multiple instrumental variables and a single instrumental variable respectively) from the TwoSampleMR package[59,60] 'https://github.com/mrcieu/TwoSampleMR'. We adapted the MR-PCA and MR-IVW-LD code from Burgess and Thompson[23] our adaptation was limited to storing duplicated code segments in memory that can otherwise take a long while to process. (Code reference https://github.com/adriaan-vd-graaf/mrlink2, using python version 3.11.6). We compared our adaptation to the original and found no difference in effect estimates or levels of significance (Table 1). MR-IVW-LD accepts the same variants as instrumental variables; however, the method adjusts their effect sizes based on the LD in between the instrumental variables.

When determining detection rates in our simulations, we consider it evidence for a causal relationship if the MR *P* value is smaller than 0.05. We have not compared the original MR-link (v1) method as our analysis depends exclusively on summary statistics and as such, the method is not suited for our comparisons[20].

Additionally, we have applied MR-APSS (v0.0.0.9000) and MR-PRESSO (v1.0) causal estimates in appropriate comparisons in this study. Using the 'MRAPSS' and 'MRPRESSO' R packages, we clumped ($r^2 < 0.001$, 1000Kb window) the summary statistics at $P \leq 5 \times 10^{-8}$ for MR-PRESSO (genome wide significance), and $P \leq 5 \times 10^{-8}$ for MR-APSS, using functionality provided by the MR-APSS package, following recommendations by the MR-APSS authors.

**The MR-link-2 likelihood function.** MR-link-2 is a likelihood function that models the summary statistics found between a cohort of $n_X$ exposure phenotypes ($X$) and an $n_Y$ outcome phenotypes ($Y$). The full derivation of the MR-link-2 likelihood function can be found in the Supplementary Note. We continue with a bird's eye view of the full derivation.

We model the causal relationship $\alpha$ between an exposure and outcome in the following way:

$$\mathbf{X} = \mathbf{G} \cdot \boldsymbol{\gamma}^{(\mathbf{X})} + \boldsymbol{\epsilon}_\mathbf{X} \tag{1}$$

$$\mathbf{Y} = \alpha \cdot \mathbf{X} + \mathbf{G} \cdot \boldsymbol{\gamma}^{(\mathbf{Y})} + \boldsymbol{\epsilon}_\mathbf{Y} \tag{2}$$

Here, $\mathbf{G}$ represents a genotype matrix, with normalized genotypes to zero-mean and unit variance across samples. SNP effects are modelled as random, $\boldsymbol{\gamma}^{(\mathbf{X})} \sim N(0, \sigma_\mathbf{X}^2)$ and $\boldsymbol{\gamma}^{(\mathbf{Y})} \sim N(0, \sigma_\mathbf{Y}^2)$, where $\sigma_\mathbf{X}^2$ is a per variant heritability estimate related to the *cis* heritability of $\mathbf{X}$, such that $\sigma_\mathbf{X}^2 = h_\mathbf{X}^2/m$ and $\sigma_\mathbf{Y}^2$ is the per variant direct (vertically pleiotropic) *cis* heritability of $\mathbf{Y}$, such that $\sigma_\mathbf{Y}^2 = h_\mathbf{Y}^2/m$. Error terms are distributed as follows: $\boldsymbol{\epsilon}_\mathbf{X} \sim N(0, 1 - h_\mathbf{X}^2)$ and $\boldsymbol{\epsilon}_\mathbf{Y} \sim N(0, 1 - \alpha^2 - h_\mathbf{Y}^2)$. The marginal GWAS summary statistics, estimated in samples of size $n_\mathbf{X}$ and $n_\mathbf{Y}$, respectively, can then be written as:

$$\widehat{\boldsymbol{\beta}}_\mathbf{X} = \mathbf{C} \cdot \boldsymbol{\gamma}^{(\mathbf{X})} + \boldsymbol{\eta}_\mathbf{X} \tag{3}$$

$$\widehat{\boldsymbol{\beta}}_Y = \mathbf{C} \cdot \left(\alpha \cdot \boldsymbol{\gamma}^{(\mathbf{X})} + \boldsymbol{\gamma}^{(\mathbf{Y})}\right) + \boldsymbol{\eta}_\mathbf{Y} \tag{4}$$

Where $\mathbf{C}$ represents an $m$ by $m$ LD matrix: $\mathbf{C} = \mathbf{G}^\mathbf{T} \cdot \mathbf{G}/n$ and $\boldsymbol{\eta}_\mathbf{X} = \alpha \cdot (\mathbf{G}^\mathbf{T} \cdot \boldsymbol{\epsilon}_\mathbf{X})/n_\mathbf{X}$ which is distributed as $\sim N\left(0, \mathbf{C} \cdot \frac{(1 - h_\mathbf{X}^2)}{n_\mathbf{X}}\right)$. $\boldsymbol{\eta}_\mathbf{Y} = \alpha \cdot (\mathbf{G}^T \cdot \boldsymbol{\epsilon}_\mathbf{X})/n_\mathbf{Y} + (\mathbf{G}^T \cdot \boldsymbol{\epsilon}_\mathbf{Y})/n_\mathbf{Y}$ is distributed as $\sim N\left(0, \mathbf{C} \cdot \frac{(1 - \alpha^2 \cdot h_\mathbf{X}^2 - h_\mathbf{Y}^2)}{n_\mathbf{X}}\right)$.

These distributional assumptions allow us to formulate a joint likelihood function for both summary statistics to estimate the causal

effect and underlying multivariable SNP effects: $L(\hat{\boldsymbol{\beta}}_{\mathbf{X}}, \hat{\boldsymbol{\beta}}_{\mathbf{Y}} | \alpha, \boldsymbol{\gamma}_{\mathbf{X}}, \boldsymbol{\gamma}_{\mathbf{Y}})$. To maximize the likelihood function, we would have to optimize $2 \cdot m + 1$ variables which can be difficult, as $m$ could contain thousands of parameters, when an associated region has many genetic variants. Therefore, we integrate out the underlying SNP effects $\boldsymbol{\gamma}^{(\mathbf{X})}$ and $\boldsymbol{\gamma}^{(\mathbf{Y})}$, conditional on the per SNP heritabilities $\sigma_{\mathbf{Y}}^2$ and $\sigma_{\mathbf{X}}^2$. This reduces the parameters to optimize to 3, which is faster and increases power. After some algebraic transformations (Supplementary Note), the MR-link-2 log likelihood function simplifies to:

$$
\begin{aligned}
L(\hat{\boldsymbol{\beta}}_{\mathbf{X}}, \hat{\boldsymbol{\beta}}_{\mathbf{Y}} | \alpha, \sigma_{\mathbf{X}}^2, \sigma_{\mathbf{Y}}^2) = & -m \cdot \log(2\pi) \\
& -\frac{1}{2} \cdot \sum_{i=1}^{m} \log\left( (\alpha^2 \cdot n_{\mathbf{Y}} + n_{\mathbf{X}}) \cdot \lambda_i + \sigma_{\mathbf{X}}^{-2} - \frac{\alpha^2 \cdot n_{\mathbf{Y}}^2 \cdot \lambda_i^2}{n_{\mathbf{Y}} \cdot \lambda_i + \sigma_{\mathbf{Y}}^{-2}} \right) \\
& -\frac{1}{2} \cdot \sum_{i=1}^{m} \log(n_{\mathbf{Y}} \cdot \lambda_i + \sigma_{\mathbf{Y}}^{-2}) \\
& +\frac{1}{2} \sum_{i=1}^{m} \left( \left(\hat{\boldsymbol{\delta}}^{(\mathbf{X})}\right)_i^2 \cdot D_{(i,i)}^{(\mathbf{X},\mathbf{X})} + \sum_{i=1}^{m} \hat{\boldsymbol{\delta}}_i^{(\mathbf{X})} \cdot \hat{\boldsymbol{\delta}}_i^{(\mathbf{Y})} \cdot D_{(i,i)}^{(\mathbf{X},\mathbf{Y})} \right) \\
& +\frac{1}{2} \sum_{i=1}^{m} \left( \left(\hat{\boldsymbol{\delta}}^{(\mathbf{Y})}\right)_i^2 \cdot D_{(i,i)}^{(\mathbf{Y},\mathbf{Y})} \right) \\
& -\frac{n_{\mathbf{X}}}{2} \sum_{i=1}^{m} \left( \left(\hat{\boldsymbol{\delta}}_i^{(\mathbf{X})}\right)^2 / \lambda_i \right) - \frac{n_{\mathbf{Y}}}{2} \sum_{i=1}^{m} \left( \left(\hat{\boldsymbol{\delta}}_i^{(\mathbf{Y})}\right)^2 / \lambda_i \right) \\
& +\frac{m}{2} \cdot (\log(n_{\mathbf{X}}) + \log(n_{\mathbf{Y}})) \\
& -\sum_{i=1}^{m} \log(\lambda_i) - m \cdot (\log(\sigma_{\mathbf{X}}) + \log(\sigma_{\mathbf{Y}}))
\end{aligned}
$$

(5)

To arrive to this expression, we used a singular value decomposition of the correlation matrix, $\mathbf{C} = \mathbf{U} \cdot \boldsymbol{\Lambda} \cdot \mathbf{U}^T$, which preserved 99% of the variance. This led to the introduction of the following quantities: $\lambda_i$ is the $i^{\text{th}}$ diagonal element of $\boldsymbol{\Lambda}$, $\hat{\boldsymbol{\delta}}^{(\mathbf{X})} = \mathbf{U}^T \cdot \hat{\boldsymbol{\beta}}^{(\mathbf{X})}$ and $\hat{\boldsymbol{\delta}}^{(\mathbf{Y})} = \mathbf{U}^T \cdot \hat{\boldsymbol{\beta}}^{(\mathbf{Y})}$. Finally, the $\mathbf{D}^{(\mathbf{X},\mathbf{X})}$, $\mathbf{D}^{(\mathbf{X},\mathbf{Y})}$ and $\mathbf{D}^{(\mathbf{Y},\mathbf{Y})}$ are diagonal matrices with diagonal elements defined as $D_{(i,i)}^{(\mathbf{X},\mathbf{X})} = \left( ((\alpha^2 \cdot n_{\mathbf{Y}} + n_{\mathbf{X}}) \cdot \lambda_i + \sigma_{\mathbf{X}}^{-2}) - \frac{\alpha^2 \cdot n_{\mathbf{Y}}^2 \cdot \lambda_i^2}{n_{\mathbf{Y}} \cdot \lambda_i + \sigma_{\mathbf{Y}}^{-2}} \right)^{-1}$, $D_{(i,i)}^{(\mathbf{X},\mathbf{Y})} = -D_{(i,i)}^{(\mathbf{X},\mathbf{X})} \cdot \frac{\alpha \cdot n_{\mathbf{Y}} \cdot \lambda_i}{n_{\mathbf{Y}} \cdot \lambda_i + \sigma_{\mathbf{Y}}^{-2}}$ and $D_{(i,i)}^{(\mathbf{Y},\mathbf{Y})} = \frac{1}{n_{\mathbf{Y}} \cdot \lambda_i + \sigma_{\mathbf{Y}}^{-2}} + D_{(i,i)}^{(\mathbf{X},\mathbf{X})} \cdot \frac{\alpha^2 \cdot n_{\mathbf{Y}}^2 \cdot \lambda_i^2}{(n_{\mathbf{Y}} \cdot \lambda_i + \sigma_{\mathbf{Y}}^{-2})^2}$.

The full derivation and Python implementation details can be found in the (Supplementary Note).

**Application of the MR-link-2 likelihood function.** We optimize the MR-link-2 likelihood function using the Nelder-Mead optimizer using the 'scipy optimize minimize' function (scipy v1.14.1)[61]. We optimize the likelihood function three times, first by setting i) $\alpha = 0$ and freely estimating $\hat{\sigma}_{\mathbf{X}}^2$ and $\hat{\sigma}_{\mathbf{Y}}^2$. Then, ii) by setting the pleiotropic variance $\sigma_{\mathbf{Y}}^2$ to zero and freely estimating $\hat{\alpha}$ and $\hat{\sigma}_{\mathbf{X}}^2$. And finally, iii) by freely estimating all three parameters $\hat{\alpha}$, $\hat{\sigma}_{\mathbf{X}}^2$ and $\hat{\sigma}_{\mathbf{Y}}^2$. We identify confidence intervals and $P$ values of $\hat{\alpha}$, and $\hat{\sigma}_{\mathbf{Y}}^2$ through a likelihood ratio test with one degree of freedom.

We estimate that under the following assumptions, the MR-link-2 model provides accurate estimates of causality

1. The relevance assumption (IV-1): The genetic variant $\mathbf{G}$ is associated to the exposure $\mathbf{X}$
2. The independence assumption (IV-2): The genetic variant $\mathbf{G}$ is independent of any confounder $\mathbf{C}$
3. The infinitesimal genetic architecture assumption: All genetic variants $\mathbf{G}$ in a locus have a non-zero genetic effect on the exposure with zero mean
4. Independence of pleiotropy assumption: The pleiotropic effect magnitude $\mathbf{G} \rightarrow \mathbf{Y}$ are independent from the direct effects of $\mathbf{G} \rightarrow \mathbf{X}$
5. The correlation between genetic variants is measured without error

A full implementation for MR-link-2 is available online at https://github.com/adriaan-vd-graaf/mrlink2. This implementation accepts 2 harmonized summary statistic files and a plink style ".bed" genotype file used for generating an LD reference[62]. For all associated regions in the exposure summary statistics file, MR-link-2 provides a causal estimate.

## Simulations

**Simulations of summary statistics.** We performed extensive simulations to ensure that MR-link-2 provides accurate causal inference, as well as to compare it to other *cis* methods. Our simulations were performed with the goal of mimicking a *cis* region of a molecular -omics study that is potentially causal to a complex trait that is measured in a large cohort. Therefore, the exposure is measured in 10,000 individuals ($n_{\mathbf{X}}$), while the outcome is measured in 300,000 individuals ($n_{\mathbf{Y}}$) in a genomic region of 2068 genetic variants ($m$) that is derived from a UK10K region on chromosome 10[22]. Our simulations contain six different parameters that we vary: the causal effect ($\alpha \in \{0, 0.05, 0.1, 0.2\}$), the exposure heritability ($h_{\mathbf{X}}^2 \in \{0.001, 0.01, 0.1\}$), pleiotropy that is represented as outcome heritability ($h_{\mathbf{Y}}^2 \in \left\{ 10^{-20}, 10^{-5}, 10^{-4}, 0.001, 0.01, 0.03 \right\}$), The size of the linkage disequilibrium (LD) reference ($n_{ref} \in \{500, 5000, \infty\}$), the number of underlying causal SNPs ($m_{causal} \in \{1, 3, 5, 10, 100\}$) which represents a subset of $m$, the minimum and maximum LD between causal and pleiotropic SNPs ($r_{causal}^2 \in \{(0.1, 0.95), (0.01, 0.95), (0.0, 1.0)\}$) for minimum and maximum correlation respectively. In total we have simulated 3,240 different scenarios with 1000 replications per scenario. Of note, none of the 3240 parametrizations of our simulations do not violate the specific assumptions underlying MR-link-2. MR-link-2's underlying assumptions are violated when $n_{ref} \neq \infty$, $m_{causal} \neq 2068 = m$ and when $r_{causal}^2 \neq 0$.

We simulated summary statistics for the two phenotypes in the following way:

1. From the total number of markers $m$, a subset of $m_{causal}$ SNPs are selected from the region for the exposure and the outcome. Selection is random across the region when $r_{causal} = (0, 1)$ and following the procedure of the original MR-link manuscript otherwise[20]. In this procedure, SNPs are selected iteratively until $m_{causal}$ SNPs are selected. First, a SNP causal to the exposure is drawn randomly from the region, then the next exposure SNP is drawn from all possible SNPs that meet the correlation criteria compared to the previously selected SNPs. The SNPs causal to the outcome will be selected to be within the LD window of at least one exposure causal SNP.
2. Independent SNP effects for the exposure and the outcome ($\boldsymbol{\gamma}_{\mathbf{X}}$ and $\boldsymbol{\gamma}_{\mathbf{Y}}$ respectively) are randomly drawn from a normal distribution $\boldsymbol{\gamma}_{\mathbf{X}} \sim N(0, \sigma_{\mathbf{X}}^2)$ and $\boldsymbol{\gamma}_{\mathbf{Y}} \sim N(0, \sigma_{\mathbf{Y}}^2)$ for each SNP that is selected to be causal, otherwise it is set to zero.
3. Independent SNP effects $\boldsymbol{\gamma}_{\mathbf{Y}}$, $\boldsymbol{\gamma}_{\mathbf{X}}$ are transformed into unconditional effect sizes $\boldsymbol{\beta}$ in the following way. We multiply the independent SNP effects by the correlation matrix $\mathbf{C}$ and add measurement error term $\boldsymbol{\beta}_{\mathbf{X}} = \boldsymbol{\gamma}_{\mathbf{X}} \cdot \mathbf{C} + N(0, \mathbf{C} \cdot \frac{1 - h_{\mathbf{X}}^2}{n_{\mathbf{X}}})$ and $\boldsymbol{\beta}_{\mathbf{Y}} = \boldsymbol{\gamma}_{\mathbf{Y}} \cdot \mathbf{C} + N\left(0, \mathbf{C} \cdot \frac{1 - \alpha^2 \cdot h_{\mathbf{X}}^2 - h_{\mathbf{Y}}^2}{n_{\mathbf{Y}}}\right)$.

These simulated summary statistics are then introduced in their respective MR and colocalization algorithms. When $n_{ref}$ is infinite, the LD matrix that is used as input for the algorithms ($\hat{\mathbf{C}}$) is the same as the original ($\mathbf{C}$). When $n_{ref}$ is not infinite, we simulate imprecisely measured LD through Wishart sampling the $\mathbf{C}$ matrix[63]. In cases where $\mathbf{C}$ is not positive semidefinite, we add regularization constants (up to 0.5) to the diagonal of the original matrix to ensure that Wishart sampling continues correctly.

For the methods that require instrument selection (MR-IVW and MR-IVW LD) we selected instruments using *P* value clumping at a *P* value threshold of $5 \times 10^{-8}$ and an LD r$^2$ squared threshold of 0.01.

## Summary statistics used in this study
### Summary statistics harmonization and associated region selection.
In this work we utilize summary statistics from a variety of different studies. We processed summary statistics of all studies in the same way: First, if necessary, we lifted over summary statistics into human chromosome build 37 using UCSCs liftover tool (https://genome.ucsc. edu/cgi-bin/hgLiftOver) combined with their chain files (https:// hgdownload.soe.ucsc.edu/downloads.html). Then, we include SNP variants that have LD information available, by overlapping the variants (based on chromosome, position and alleles) present in the summary statistics file with the variants in our LD reference (UK10K). Due to potential strand inconsistencies, palindromic SNPs were removed.

We ensured that effect size magnitudes of summary statistics are the same between studies by converting to standardized effect sizes:

$$\beta_{standardized} = \frac{z}{\sqrt{n+z^2}} \qquad (6)$$

$$se(\beta_{standardized}) = \frac{1}{\sqrt{n+z^2}} \qquad (7)$$

where the *n* is the sample size of the tested SNP and *z* is the Z score of the tested SNP. If *n* was not available per SNP, we set *n* to be the maximum sample size reported by the authors. If *z* was not available, the P value of the SNP-trait association combined with the effect direction was converted into a Z score. Genetic associations are retained if they have at least a minor allele frequency of 0.5% in the UK10K LD reference and if the variant has been measured in at least 95% of the maximum number of measured individuals (if this information was available) (Supplementary Note).

We identify associated regions using the –clump command of plink (v1.90b7)[62], using a clumping window of 250Kb, an LD threshold of 0.01 *r$^2$* and a P value threshold of $5 \cdot 10^{-8}$. If clumped regions overlap, we combine them together, so these regions can be much larger than 250Kb. All harmonized regions are then analyzed by each *cis* genetic method individually. If the method requires the selection of IVs (MR-IVW, MR-IVW LD), these are clumped inside the region at a P value threshold of $5 \cdot 10^{-8}$ and an r$^2$ threshold of 0.01.

### Summary statistics of metabolite QTL studies.
We analyzed the summary statistics of four different mQTL studies. To match the associations to our reference panel, we chose to analyze the European component of the individuals, when available. Shin et al. was downloaded from http://metabolomips.org/gwas/index.php?task= download[33], Lotta et al. was downloaded from https://omicscience. org/apps/crossplatform/[32], Chen et al. summary statistics derived from European populations were downloaded from the GWAS catalog[31,64]. The UK biobank summary statistics were downloaded from the IEU open GWAS project, where we included 19 accessions that represented small metabolites[30,65]. More information about the metabolites used in this study can be found in (Supplementary Data 4).

### Summary statistics of complex trait harmonization and processing.
To understand the behavior of *cis* MR methods, we selected ground truth (non-)causal relationships between complex trait combinations from Morrison et al.[9] These involve 10 unique phenotypes, summary statistics of which were downloaded from their respective datasets (Supplementary Data 16). In brief, we utilized summary statistics from lipid phenotypes (LDL-C, HDL-C and total cholesterol) from Graham et al.[66], blood pressure (diastolic blood pressure and systolic blood pressure) summary statistics from Warren et al.[67], coronary artery disease summary statistics from Aragam et al.[68], stroke summary statistics from Mishra et al.[69], childhood asthma summary statistics from Ferreira et al.[70], Type 2 diabetes summary statistics from Mahajan et al.[71] summary statistics of smoking from Karlsson Linnér et al.[72], circulating IL18 measurements were derived from Sun et al.[73] C-reactive protein was measurements were from Said et al.[74], COVID-19 was download from the Covid host genetics consortium (https://www. covid19hg.org/results/r7/, using the 'Very severe respiratory confirmed covid vs. population – only europeans' summary statistics)[75] and atopic dermatitis was downloaded from Budu-Aggrey et al.[76] C-reactive protein was downloaded from Said et al.[74]. Some of these studies are based on multi-ancestry analyses, when population specific summary statistics were available, we exclusively analyzed the European subset of the final summary statistics (Supplementary Data 16).

### Summary statistics of eQTLGen.
The eQTLGen Consortium is an initiative to investigate the genetic architecture of blood gene expression and to understand the genetic basis of complex traits. We used interim summary statistics from eQTLGen phase 2, wherein a genome-wide eQTL analysis has been performed in 19 cohorts, representing 14,855 individuals. Numbers of individuals and their sex can be found in Supplementary Data 17. Of note, all cohorts except the INTERVAL[77] cohort were part of the original publication, thus the INTERVAL cohort is a new addition to the consortium (Supplementary Note).

All 19 cohorts performed cohort-specific analyses as outlined in the eQTLGen analysis cookbook (https://eqtlgen.github.io/eqtlgen-web-site/eQTLGen-p2-cookbook.html). Genotype quality control was performed according to standard bioinformatics practices and included quality metric-based variant and sample filtering, removing related samples, ethnic outliers and population outliers. Genotype data was converted to genome build hg38 if not done so already and the autosomes were imputed using the 1000 G 30x WGS reference panel[78] (all ancestries) using our imputation pipeline (https://github.com/ eQTLGen/eQTLGenImpute). Like the genotype data, gene expression data was processed using our data QC pipeline (https://github.com/ eQTLGen/DataQC). For array-based datasets, we used the results from empirical probe mapping approach from our previous study[21] to connect the most suitable probe to each gene which has previously been to show expression in the combined BIOS whole blood expression dataset. Raw expression data was further normalized in accordance with the expression platform used (quantile normalization for Illumina expression arrays and TMM[79] for RNA-seq) and inverse normal transformation was performed. Gene expression outlier samples were removed and gene summary information was collected for filtering at the central site. Samples for whom there were mismatches in genetically inferred sex, reported sex, or the expression of genes encoded from sex chromosomes were removed. Similarly, samples with unclear sex, based on genetics or gene expression were removed.

The HASE framework (using Python version 2.7.15)[80] was used to perform genome-wide meta-analysis. For genome-wide eQTL analysis, this limits the data transfer size while ensuring participant privacy. At each of the cohorts, the quality controlled and imputed data was processed and encoded so that the individual level data can no longer be extracted, but while still allowing effect sizes to be calculated for the linear relationship between variants and genes. (https://github.com/eQTLGen/ConvertVcf2Hdf5 and https://github. com/eQTLGen/PerCohortDataPreparations).

Centrally, the meta-analysis pipeline was run on the 19 cohorts. The pipeline which performs per cohort calculations of effect sizes and standard errors and the inverse variance meta-analysis is available at https://github.com/eQTLGen/MetaAnalysis (using Python 3.11.3). We included 4 genetic principal components as covariates. Per every dataset, genes were included if the fraction of unique expression

values was equal or greater than 0.8, Variants were included based on imputation quality, Hardy-Weinberg equilibrium and minor allele frequency (MAF) (Mach $R^2 \geq 0.4$, Hardy-Weinberg $P \geq 1 \times 10^{-6}$ and MAF $\geq 0.01$).

**Summary statistics of cell type proportions.** We processed the summary statistics of 15 cell type composition phenotypes from the Chen et al. (2020) meta-analysis, using the summary statistics of individuals with European ancestry. these cell type composition phenotypes were downloaded from http://www.mhi-humangenetics.org/en/resources/[51,52].

**Harmonization of metabolites.** The mQTL studies studied here use different metabolite naming schemes for their metabolites. To make sure that all metabolites studied are the same, we harmonized metabolites to HMDB database identifiers[36]. The HMDB is a large reference database for human metabolites and contains references to other metabolites. In this work we utilized the HMDB database of 17th of November 2022. Downloaded from https://hmdb.ca/system/downloads/current/hmdb_metabolites.zip.

As a starting point for the harmonization of metabolites, we utilized the metabolite comparison information provided by the supplementary table 4 of the Chen et al. publication[31]. Here, 2075 metabolites across 6 studies were provided a harmonized name. One mQTL study used in this work was not considered (UKB metabolites), therefore, we manually harmonized 19 metabolites from the unharmonized UKB information into a derived table (Supplementary Data 4). Combined, the 4 mQTL studies under investigation have 1518 unique harmonized metabolites measured. We removed 430 measured metabolites that were not matchable to a single compound, leaving 1035 metabolites for study. 660/1035 metabolites were already matched to the HMDB database by Chen et al. For the remaining matches, we matched name and synonyms for a further 239 matched names to HMDB identifiers. As a final step we manually matched a further 56 compounds based on manual web searching the HMDB website, resulting in 854 metabolites with a HMDB identifier (Supplementary Data 4). To ensure that metabolites can be easily matched with other databases, these metabolites have also been matched with InChIKey, KEGG compound ID and ChEBI ID (Supplementary Data 4).

**Metabolite networks used in this study.** We created 3 different metabolite networks to benchmark our causal inference method: the MetaCyc HumanCyc v24 pathway (released 30th of April 2020), the WikiPathways Homo sapiens ("https://wikipathways-data.wmcloud.org/current/gpml/wikipathways-20230510-gpml-Homo_sapiens.zip") pathway and the KEGG pathway (downloaded on the 30th of May 2023). Each pathway has an extended graph and a measured graph. The measured graph contains the direct reactions of the measured metabolites in this study, whereas the extended graph contains all the causal relationships in the pathway definition.

**Creation of the MetaCyc network graph.** To create the MetaCyc metabolite network we loaded the compound and reaction information from downloaded MetaCyc HumanCyc flat files[27]. Some compounds are very common reactants, therefore, we removed the following HumanCyc Identifiers from our analysis: {'GTP', 'CL-', 'CYS', 'Fatty-Acids', 'HCO3', 'GDP', '3-5-ADP', 'MALONYL-COA','NADH-P-OR-NOP', 'CMP', 'PAPS', 'NAD-P-OR-NOP', 'SUC', 'Acceptor', 'AMMONIUM', 'NA + ', 'ACETYL-COA', 'ADENOSYL-HOMO-CYS', 'HYDROGEN-PEROXIDE','UDP', 'AMP', 'Donor-H2','NADH', 'PPI', 'NADPH', 'ADP', 'NAD', 'CARBON-DIOXIDE', 'Pi', 'CO-A', 'NADP', 'ATP','OXYGEN-MOLECULE', 'WATER', 'PROTON'}. After common reactant removal, we built an extended graph containing 2,560 compounds and 3,900 reactions. We

matched HumanCyc compound identifiers with HMDB and ChEBI identifiers[27,81]. After matching with our mQTL studies, this resulted in 115 compounds across 146 reactions in the measured graph.

**Creation of the WikiPathways network graph.** To create the WikiPathways metabolite network, we downloaded each individual human pathway and kept all combinations where a compound is converted into another according to the 'mim-conversion' arrow specifier[29]. This resulted in an extended graph containing 3795 compounds and 4871 reactions. We matched the compounds in WikiPathways with the HMDB[36] or the ChEBI databases[81] and find 160 compounds across 155 reactions that were measured in the measured graph.

**Creation of the KEGG network graph.** To identify the KEGG metabolite network, we used the following procedure. For the 435 compounds for which a KEGG ID was matched, we downloaded the compound data and determined in which full pathways ('map') the compound could be found. For each of these 229 pathways, we downloaded the human equivalent (replacing 'map' with 'hsa') KGML files. From these KGML files, all human reactions were parsed to construct a graph of 1877 reactions across 1270 compounds in the extended graph. 113 of which were measured in at least one mQTL study. This measured graph contains 126 reactions.

**Bias estimation and le Chatelliers principle in mQTL MR.** We test the bias of MR methods by comparing causal estimates of the same metabolite when they are measured in different studies (Fig. 4a–d). We take the mean of the effect sizes of these 'self comparisons' when the respective MR method identifies the causal estimate as nominally significant.

We can determine if the causal estimates of the mQTL analyses seem to represent metabolism by ensuring that the causal estimate is positive, which would represent a chemical reaction between a substrate and a product in equilibrium conditions. We test for this "*le Chatelliers principle*" By taking the proportion of positive causal estimates compared to the total number of Bonferroni significant MR estimates (Fig. 4e–h) (Supplementary Note).

**Reference set of metabolite reactions.** To benchmark the *cis* genetic methods used in this publication, we define a ground truth set of reactions. Real reactions are defined when the substrate 'causes' changes in the levels of the product. Unfortunately, it is difficult to define false causal links, as it is almost impossible to prove that two metabolites are not in a reaction together. On top of that, it could be that a metabolite is understudied and therefore a potential reaction is simply not known. Our approach defines negative metabolite combinations based on distance in the graph of "ground truth" reactions. If the minimum distance (counted as reactions) between metabolites is a certain number of steps or more, we consider the combination as a ground truth null reaction, as it is unlikely that any statistical method will have the power to pick up a signal after a certain number of reactions. We do not consider metabolite combinations a false causal link if there is no path possible between them in the extended graph for two reasons. First, we reduce the chance of an understudied metabolite being considered non-causal as they are present in the metabolite network that we test. Secondly, this approach ensures that the exposures studied are used both for true causal link and false causal links datasets, making the analysis less dependent on which associated regions are used, as they are the same or similar between all sets. Of note, sometimes a causal reaction is bidirectional, which translates to both forward and backward causal links being considered as ground truth.

**Meta-analyzing multiple associated regions.** When there are multiple associated gene regions available for a metabolite, it becomes

possible to meta-analyze regions. We meta-analyze regions by taking the weighted mean $\bar{\alpha}$ (and standard error se($\bar{\alpha}$)) of all associated regions together:

$$\bar{\alpha} = \left( \frac{\sum_{r=1}^{k} \frac{\hat{\alpha}_r}{\text{se}(\hat{\alpha}_r)^2}}{\sum_{r=1}^{k} \text{se}(\hat{\alpha}_r)^{-2}} \right), \qquad (8)$$

$$\text{se}(\bar{\alpha}) = \sqrt{\left( \sum_{r=1}^{k} \text{se}(\hat{\alpha}_r)^{-2} \right)} \qquad (9)$$

where $k$ are all the associated regions found in the initial clumping step, $\hat{\alpha}_r$ is the regional estimate with its associated standard error: se($\hat{\alpha}_r$).

### Analysis of complex traits

**MR detection rates and false positive rates canonical causality.** Next to metabolites, we turn to canonical causal relationships between complex traits. We perform regional estimates to understand the per region false positive rate and detection rate of MR methods (Fig. 5a-b). For each regional causal estimate we determine if it is nominally significant (0.05) and report the proportion of causal.

**Heterogeneity of causal estimates.** We meta-analyze the causal estimates of canonical causality in the same way as the metabolites (**Methods**). We estimate Cochran's Q statistics across all these associated regions by taking the sum of the Z score deviation of a regional estimate with the meta-analyzed estimate for each trait pair (Fig. 5c).

### Analysis of gene expression and cell types

We used cell type definitions and their marker genes to determine if MR methods correctly identify the causal direction between cell types and their marker genes. Marker genes were taken from the Azimuth PBMC cell type reference (http://www.mhi-humangenetics.org/en/resources/)[54], which is typically used to identify cell types from single cell RNA sequencing experiments. We can identify marker genes for 10 out of 15 cell types. For each cell type composition / marker gene combination (41 in total), we perform bidirectional MR between cell types and their respective marker genes, where we assume that the marker genes are the cause of the cell type and not vice versa (Supplementary Data 15) (Supplementary Data 17) (Fig. 5e). We combine all cell type–marker gene causal relationships together and use $P$ values to determine discriminative ability in terms of AUC (Fig. 5f).

### Reporting summary

Further information on research design is available in the Nature Portfolio Reporting Summary linked to this article.

## Data availability

The summary statistics for the metabolite analysis, the complex trait analysis and the blood cell type composition phenotypes are available from the respective source publications (Supplementary Data 16).

The data used from the eQTLGen Consortium cohorts is listed in the (Supplementary Note) and Võsa et al.[21]. Per-cohort summary statistics for discovery cohorts can be made available after approval of an analysis proposal (https://eqtlgen.github.io//eqtlgen-web-site/documents.html) in eQTLGen and with agreement of the cohort PIs.

This Phase 2 analysis of the eQTLGen Consortium uses previously published data from the INTERVAL study[77], this data is available to bona fide researchers from ceudataaccess@medschl.cam.ac.uk. The data access policy for the data is available at http://www.donorhealth-btru.nihr.ac.uk/project/bioresource. The RNA-sequencing data are available at the European Genome-phenome Archive (EGA) under the accession number EGAD00001008015.

The newly generated eQTLGen Consortium summary statistics can be found under accession https://doi.org/10.5281/zenodo.14982207.

The genotype information underlying the LD matrices for the UK10K data resource were downloaded from the EGA under accession IDs EGAD00001000740 and EGAD00001000741. As this is individual level genotype data, a data access agreement is required for access. Conditions for this data access agreement can be found at https://www.uk10k.org/data_access.html.

## Code availability

The code for simulations, working examples and the true positives used in this study for MR-link-2 and the other *cis-* causal inference methods are available at: https://github.com/adriaan-vd-graaf/mrlink2, and https://doi.org/10.5281/zenodo.14961110

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

## Acknowledgements

We would like to thank all the study participants for their altruistic donations of their biological materials. For the acknowledgments to the cohorts of the eQTLGen Consortium, we refer to the Supplementary Note. MCB's contribution to this work was supported by the UK Medical Research Council [grant number MC_UU_00032/5]. L.F. is supported by a grant from the Dutch Research Council (ZonMW-VICI 09150182010019 to L.F.), and a sponsored research collaboration with Biogen and Roche. This work is co-financed by Oncode Institute, which is partly funded by the Dutch Cancer Society. This project has received funding from the European Union's Horizon Europe research and innovation programme under grant agreement No 101057553, a grant from Oncode Accelerator and a grant from Saxum Volutum (Pericode). Z.K. was funded by the Swiss National Science Foundation (SNSF 315230-219587).

## Author contributions

Conceptualization: Z.K., Avd.G. Methodology: Z.K., Avd.G., R.W., U.V., L.F., M.C.B., E.C. Investigation: Z.K., Avd.G., R.W., U.V., L.F. Visualization: Avd.G. Funding acquisition: Z.K., L.F., M.C.B. Project administration: Z.K. Supervision: Z.K. Writing –original draft: Avd.G. Writing–review & editing: C.A., M.C.B., Z.K., L.H., R.W., U.V., Avd.G.

## Competing interests

The main authors of this study do not declare a competing interest. The authors of the eQTLGen consortium declare the following competing interests: B.M.P. serves on the Steering Committee for the Yale Open Data Access Project funded by Johnson & Johnson. This activity is unrelated to this work. M.I. is a trustee of the Public Health Genomics (PHG) Foundation, a member of the Scientific Advisory Board of Open Targets, and has a research collaboration with AstraZeneca that is unrelated to this study. D.S.P. is an employee and stockholder of AstraZeneca. The other authors of the eQTLGen consortium do not declare competing interests.

## Additional information

# eQTLGen Consortium

Toni Boltz[10], Dorret I. Boomsma[11,12,13], Andrew Brown[14], Evans Cheruiyot[15], Emma E. Davenport[16], Théo Dupuis[14], Tõnu Esko[6], Aiman Farzeen[17,18,19], Luigi Ferrucci[20], Lude Franke ®[3,4], Timothy M. Frayling[21], Greg Gibson[22], Christian Gieger[18,19], Marleen van Greevenbroek[23], Binisha Hamal Mishra[24,25,26], M. Arfan Ikram[27], Michael Inouye[28,29,30,31,32,33], Rick Jansen[12,34], Mika Kähönen[25,35], Viktorija Kukushkina[6], Sandra Lapinska[36], Terho Lehtimäki[24,25,26], Reedik Mägi[6], Angel Martinez-Perez[37], Allan F. McRae[15], Joyce van Meurs[38,39], Lili Milani[6], Grant W. Montgomery[15], Sini Nagpal[22], Matthias Nauck[40], Roel Ophoff[10,36,41], Bogdan Pasaniuc[10,36,42,43], Dirk S. Paul[28,29,44], Elodie Persyn[28,29,30], Annette Peters[19,45,46,47], Holger Prokisch[17,48,49], Olli T. Raitakari[50,51,52,53], Emma Raitoharju[54,55], Andrew Singleton[56], Eline Slagboom[57], José Manuel Soria[37], Juan Carlos Souto[37], Alexander Teumer[58,59,60], Alex Tokolyi[16], Jan Veldink[61], Joost Verlouw[38], Ana Viñuela[62], Peter M. Visscher[15], Uwe Völker[59,63], Urmo Võsa[6], Robert Warmerdam ®[3,4], Stefan Weiss[63], Harm-Jan Westra[3,4], Andrew R. Wood[64] & Manke Xie[22]

[10]Department of Human Genetics, David Geffen School of Medicine, University of California Los Angeles, Los Angeles, CA, USA. [11]Amsterdam Reproduction & Development (AR&D) research institute, Amsterdam, the Netherlands. [12]Amsterdam Public Health research institute, Amsterdam, the Netherlands. [13]Department of Complex Trait Genetics, Center for Neurogenomics and Cognitive Research, Vrije Universiteit Amsterdam, Amsterdam, The Netherlands. [14]Population Health and Genomics, University of Dundee, Dundee, Scotland, UK. [15]Institute for Molecular Bioscience, The University of Queensland, Brisbane, Australia. [16]Wellcome Sanger Institute, Wellcome Genome Campus, Hinxton, UK. [17]Institute of Neurogenomics, Computational Health Center, Helmholtz Munich, Neuherberg, Germany. [18]Research Unit of Molecular Epidemiology, Helmholtz Zentrum München - German Research Center for Environmental Health, Neuherberg, Germany. [19]Institute of Epidemiology, Helmholtz Zentrum München - German Research Center for Environmental Health, Neuherberg, Germany. [20]Translational Gerontology Branch, National Institute on Aging, National Institutes of Health, Baltimore, MD, USA. [21]Department of Genetic Medicine and Development, CMU, University of Geneva, Geneva, Switzerland. [22]Center for Integrative Genomics, Georgia Institute of Technology, Atlanta, GA, USA. [23]CARIM, Maastricht University, Maastricht, The Netherlands. [24]Department of Clinical Chemistry, Faculty of Medicine and Health Technology, Tampere University, Tampere, Finland. [25]Finnish Cardiovascular Research Center Tampere, Faculty of Medicine and Health Technology, Tampere University, Tampere, Finland. [26]Department of Clinical Chemistry, Fimlab Laboratories, Tampere, Finland. [27]Department of Epidemiology, Erasmus MC University Medical Center, Rotterdam, The Netherlands. [28]British Heart Foundation Cardiovascular Epidemiology Unit, Department of Public Health and Primary Care, University of Cambridge, Cambridge, UK. [29]Victor Phillip Dahdaleh Heart and Lung Research Institute, University of Cambridge, Cambridge, UK. [30]Cambridge Baker Systems Genomics Initiative, Department of Public Health and Primary Care, University of Cambridge, Cambridge, UK. [31]Cambridge Baker Systems Genomics Initiative, Baker Heart and Diabetes Institute, Melbourne, VIC, Australia. [32]Health Data Research UK Cambridge, Wellcome Genome Campus and University of Cambridge, Cambridge, UK. [33]British Heart Foundation Centre of Research Excellence, University of Cambridge, Cambridge, UK. [34]Amsterdam UMC location Vrije Universiteit Amsterdam, Department of Psychiatry & Amsterdam Neuroscience -Complex Trait Genetics (VUmc) and Mood, Anxiety, Psychosis, Stress & Sleep, Amsterdam, The Netherlands. [35]Department of Clinical Physiology, Tampere University Hospital, Tampere, Finland. [36]Bioinformatics Interdepartmental Program, University of California Los Angeles, Los Angeles, CA, USA. [37]Unit of Genomic of Complex Diseases, Institut de Recerca Sant Pau (IR Sant Pau), Barcelona, Spain. [38]Department of Internal Medicine, Erasmus MC University Medical Center, Rotterdam, The Netherlands. [39]Department of Orthopaedics and Sportsmedicine, Erasmus MC University Medical Center, Rotterdam, The Netherlands. [40]Institute of Clinical Chemistry and Laboratory Medicine, University Medicine Greifswald, Greifswald, Germany. [41]Center for Neurobehavioral Genetics, Semel Institute for Neuroscience and Human Behavior, David Geffen School of Medicine, University of California Los Angeles, Los Angeles, USA. [42]Department of Computational Medicine, David Geffen School of Medicine, University of California Los Angeles, Los Angeles, CA, USA. [43]Department of Pathology and Laboratory Medicine, David Geffen School of Medicine, University of California Los Angeles, Los Angeles, CA, USA. [44]Centre for Genomics Research, Discovery Sciences, BioPharmaceuticals R&D, AstraZeneca, Cambridge, UK. [45]Chair of Epidemiology, IBE, Faculty of Medicine, LMU Munich, Munich, Germany. [46]German Centre for Cardiovascular Research (DZHK), Partner Site Munich Heart Alliance, Munich, Germany. [47]German Center for Diabetes Research (DZD), Neuherberg, Germany. [48]School of Medicine, Institute of Human Genetics, Technical University of Munich, Munich, Germany. [49]German Center for Child and Adolescent Health (DZKJ), partner site Munich, Munich, Germany. [50]Research centre of Applied and Preventive Cardiovascular Medicine, University of Turku, Turku, Finland. [51]Department of Clinical Physiology and Nuclear Medicine, Turku University Hospital, Turku, Finland. [52]Centre for Population Health Research, University of Turku and Turku University Hospital, Turku, Finland. [53]InFLAMES Research Flagship, University of Turku, Turku, Finland. [54]Molecular Epidemiology, Faculty of Medicine and Health Technology, Tampere University, Tampere, Finland. [55]Tampere University Hospital, Tampere, Finland. [56]Laboratory of Neurogenetics, National Institute on Aging, National Institutes of Health, Bethesda, MD, USA. [57]Section of Molecular Epidemiology, Department of Biomedical Data Sciences, Leiden University Medical Center, Leiden, the Netherlands. [58]Department of Psychiatry and Psychotherapy, University Medicine Greifswald, Greifswald, Germany. [59]DZHK (German Center for Cardiovascular Research), Partner Site Greifswald, Greifswald, Germany. [60]Department of Population Medicine and Lifestyle Diseases Prevention, Medical University of Bialystok, Bialystok, Poland. [61]Department of Neurology, UMC Utrecht Brain Center Rudolf Magnus, Utrecht, The Netherlands. [62]Biosciences Institute, Faculty of Medical Sciences, University of Newcastle, Newcastle upon Tyne, UK. [63]Interfaculty Institute for Genetics and Functional Genomics, University Medicine Greifswald, Felix-Hausdorff-Strasse 8, Greifswald, Germany. [64]College of Medicine and Health, University of Exeter, Exeter, UK.

