## [Transparent Peer Review file · Nature Communications]

MR-link-2: pleiotropy robust cis Mendelian randomization validated in three independent reference datasets of causality

Corresponding Author: Professor Zoltan Kutalik

Version 0:

Reviewer comments:

Reviewer #1

(Remarks to the Author)

This is a potentially fantastically powerful resource and I commend the authors on a very comprehensive paper.

What is missing is a series of positive and negative control real life examples, to show how well the approach works in practice.

I would recommend the following types of examples:

Positive controls

HMGCR gene, LDLc and coronary artery disease

IL18 gene, IL18 and atopic dermatitis

IL6R gene, CRP and covid-19

Negative controls

IL6R gene, CRP and type 2 diabetes mellitus

PCSK9 gene, LDLc and BMI

CRP gene, CRP and coronary artery disease

The authors should also be clear that horizontal pleiotropy in this context exclusively refers to a variant in LD. Every colocalising signal could be considered a target effect, although we may not know in which cell or tissue it is occurring.

In the meantime, I will also practice some positive and negative controls using the details on the github page.

(Remarks on code availability)

I will have applied the code in time for considering the next iteration of the manuscript.

Reviewer #2

(Remarks to the Author)

Please find my comments to the authors in the attached pdf file "ReviewMRlink2".

(Remarks on code availability)

The software provided on GitHub is clear. It also provides a REAME file, instructions, and some examples.

Reviewer #3

(Remarks to the Author)

The authors present a new version of their previously developed method MR-link, MR-link2. The new version is meant to address single region associations. The authors report around 30% improvement in accuracy compared to selected algorithms, and present results on both simulated and an interesting example on metabolomics data.

While improvements in mendelian randomization techniques are important for addressing specific challenges, there are some major concerns as follows:

1-The proposed improvement of cis MR-link2 as it is presented here is not a significant improvement: Specifying that the current version mainly addresses the single association region is not considered a significant improvement. Also, addressing LD and pleiotropy have already been part of the authors' first version. There needs to be a clear difference between the two versions and the enhancement should be clarified in the abstract and in results.

2- Comparisons to previous methods should be more extensive as there are many sophisticated MR methods recently published, for example how does the new version perform in comparison with MR PRESSO (<https://www.nature.com/articles/s41588-018-0099-7>), MR APSS (<https://www.pnas.org/doi/full/10.1073/pnas.2106858119>), and others? Including more algorithms for comparison will be more convincing to researchers and make it more attractive to be used in the future, which is the main aim of the work.

3-The write up of the manuscript needs much improvement. For example, one cannot get clear messages from the results section as it is currently a mix of results, discussion and methods. Results should be more clear, focused and ideas organized.

Other minor concerns are attached as a pdf.

(Remarks on code availability)

Comments are included in the attached pdf.

Reviewer #4

(Remarks to the Author)

The review is attached.

"I co-reviewed this manuscript with one of the reviewers who provided the listed reports. This is part of the Nature Communications initiative to facilitate training in peer review and to provide appropriate recognition for Early Career Researchers who co-review manuscripts."

(Remarks on code availability)

The code review is represented in my attached report

Version 1:

Reviewer comments:

Reviewer #1

(Remarks to the Author)

I thank the authors for addressing my comments.

IL6R inhibitors for COVID-19 are a clinically validated positive control and the failure of the MR-link 2 method to identify this effect needs to be acknowledged as a limitation for analyses that are of direct translational relevance.

If practitioners relied on MR-link 2, they would not have identified the critical genetic evidence that led to tocilizumab being prioritized for treating covid-19 during the pandemic.

(Remarks on code availability)

Reviewer #2

(Remarks to the Author)

Thanks the authors for their efforts to address the comments I raised. I have no further comment about the revised manuscript, thanks.

(Remarks on code availability)

Reviewer #3

(Remarks to the Author)

Thanks for the detailed response with more experimental evidence and for the clarifications. There are very few minor comments and one main concern regarding the code consistency with the paper equations (a follow up on point 4.2) as pointed out in the attached document. Please clarify that as it may cause confusion between the paper and the code.

(Remarks on code availability)

Reviewer 4 has revised the code corresponding to the equations and we included that in the previous revision and this round of revisions are included in the attached document. There is an inconsistency between the paper and the part of the code that we mentioned in the last comment in the attached review (point 4.2), where we highlight the inconsistency. It is important that the authors revise this part and clarify it in the paper and code.

Reviewer #4

(Remarks to the Author)

(Remarks on code availability)

include in the pdf

We thank the reviewers for their thorough reading of the manuscript. Here, we're providing point-by-point responses to the reviewers' questions, and as a result we have thoroughly revised the manuscript.

We have also made some changes to the manuscript that were not requested by the reviewers. Specifically, We have reanalyzed all the causal inference on the metabolites, complex traits, and celltypes to eQTL analysis for two reasons. We updated the algorithm to allow the inclusion of more associated regions, and we fixed an LD matrix preprocessing bug.

First, we include more associated regions due to a change in genetic variant inclusion criteria. It used to be that we only included genetic variants that have more than 90% of measured variants across the whole summary statistics file. This is now determined at the regional level, allowing for the inclusion of more associated regions.

Second, due to a small change in the preprocessing of the linkage disequilibrium (LD) matrix to ensure that the genetic variants were always harmonized correctly, we have reanalyzed the causal inference on metabolites, complex traits, and eQTL / cell types. Due to the slight adjustments in the pre-processing step, the new manuscript has slightly different numbers, the main conclusions remain unchanged. Unfortunately due to this change in LD processing, the main biological example has reduced in significance to below the Bonferroni significant threshold. We have therefore decided to exclude this example from the manuscript. We instead replace it with a unique to MR-link-2 finding that relates to the Krebs Cycle and an example that was validated in human experiments.

Third, We have replaced the original Figure 5c panel with a new analysis, as the preprocessing of the linkage disequilibrium now provides us with robust correlations between regional Q statistics and the pleiotropy estimates of MR-link-2, which adds to the message that MR-link-2 is able to identify pleiotropy from a single region.

The simulations remain unchanged as the inconsistencies were limited to differing harmonization of real world data.

Reviewer #1 (Remarks to the Author):

This is a potentially fantastically powerful resource and I commend the authors on a very comprehensive paper.

Response 1.1:

We thank the reviewer for their kind words and we hope that the paper and the method will be useful to them. We've compiled a response to all the questions below, and hope that they will be considered favorably.

What is missing is a series of positive and negative control real life examples, to show how well the approach works in practice.

I would recommend the following types of examples:

Positive controls

HMGCR gene, LDLc and coronary artery disease

IL18 gene, IL18 and atopic dermatitis

IL6R gene, CRP and covid-19

Negative controls

IL6R gene, CRP and type 2 diabetes mellitus

PCSK9 gene, LDLc and BMI

CRP gene, CRP and coronary artery disease

Response 1.2:

We thank the reviewer for these valuable suggestions for positive control and negative controls. In the original manuscript, we applied all real-world examples that were mentioned in the MR-CAUSE paper (1). And we thank the reviewer for their suggestions of additional positive and negative controls to be tested. We interpreted the list in the

following way: we determine causality using instruments only from the region indicated by the listed gene, with the first phenotype as the exposure and the second phenotype listed as the outcome.

We downloaded summary statistics of the phenotypes that were not in the initial manuscript from the following publications: IL18 was derived from Sun et al. (2) , CRP was derived from Said et al. (3) (accession GCST90029070) and COVID-19 was download from the Covid host genetics consortium (<https://www.covid19hg.org/results/r7/> , using the 'Very severe respiratory confirmed covid vs. population -- only europeans' summary statistics) (4) and atopic dermatitis was downloaded from Budu-Aggrey et al. (5) . These summary statistics files were harmonized in the same way as described in the paper.

In the True positives (Response 1, Figure 1), all *cis* methods identify similar (positive) causal effects in all cases and almost all of them are nominally significant ($P < 0.05$), except for the case of MR-link-2 and the IL6R gene, from the CRP to covid-19 estimate by MR-link-2. We investigated this locus further and found that there is no colocalization between CRP and COVID-19 in this locus (posterior probability of a single causal SNP sharing (PP4): 0.003821) (**Supplementary Table 14**), instead coloc indicates that there are two distinct causal SNPs in the locus (PP3: 0.978306). Therefore, the available data does not provide conclusive evidence for this link and the other methods yield overly confident SEs presumably because they cannot distinguish the two separate signals. Also, note that while most of these positive controls could be identified at nominally significant levels, none of these associations would survive regional multiple testing correction ($0.05 / 104,007 = 4.8 \times 10^{-7}$), except the LDL-CAD link by MR-link-2 and MR-PCA (via *HMGCR*).

Response 1, Figure 1, the *cis* MR estimates of the True positives suggested by reviewer 1.

In the Negative controls (**Response 1, Figure 2**), MR-link-2 is the only method that identifies nominally significant effects, but this would not survive a multiple testing correction across all the regions in the complex trait analysis ($0.05 / 104,007 = 4.8 \times 10^{-7}$).

Response 1, Figure 2, the cis MR estimates of the Negative controls suggested by reviewer 1.

We have included these examples in the new manuscript. The new passage can be found between lines 469- 485 now reads (changes in red):

“

We further applied all *cis* MR methods in a “drug-target-MR” setting. MR is performed using instruments from the vicinity of one gene. As positive controls, we used the following three gene- exposure-outcome combinations: LDL-C→CAD instrumented from HMGCR, IL18→atopic dermatitis instrumented from IL18 and C-reactive protein→COVID-19 instrumented from IL6R. As negative controls, we used C-reactive protein →Type-2-diabetes 19 instrumented from IL6R, LDL-C→BMI instrumented from PCSK9 and C-reactive protein →coronary artery disease instrumented from CRP, based on established and disproven causality from these loci. Only MR-link-2 ($P = 1.2 \cdot 10^{-9}$) and MR-PCA ($P = 1.7 \cdot 10^{-7}$) identify the causal link between LDL-C and CAD instrumented from HMGCR after multiple testing correction across 104,007 pairwise regions of these complex traits ($P < 4.8 \cdot 10^{-7}$) (**Supplementary Table 14**). When using a more lenient P value threshold of only these 6 positive and negative controls ($P < 0.05 / 6$), all 3 positive controls are identified by MR-IVW LD and MR-PCA and 2 / 3 positive controls are detected by MR-link-2 and MR-IVW. This undetected positive control (C-reactive protein→COVID-19 instrumented from IL6R) has little evidence of causal variant colocalization (coloc PP4: $1.0 \cdot 10^{-16}$) in the locus, suggesting that the positive control may not be as convincing as prior studies indicate. None of the methods detect the negative controls at this lenient multiple testing threshold (**Supplementary Table 14**).

“

The authors should also be clear that horizontal pleiotropy in this context exclusively refers to a variant in LD. Every colocalising signal could be considered a target effect, although we may not know in which cell or tissue it is occurring.

Response 1.3:

The reviewer is correct to say that the MR-link-2 method is not robust to horizontal pleiotropy when the underlying causal variants of the exposure and an (unobserved

pleiotropic) phenotype are the same. This is described in the original MR-link publication (van der Graaf et al. 2020) . What we alluded to in that sentence was the situation where the pleiotropy is happening through imperfect LD (i.e. an LD-friend of the exposure instrument has a direct effect on the outcome).

In the strictest sense, when there is a single causal SNP for the exposure under investigation and this causal SNP is also causal for the pleiotropic phenotype, MR-link-2 will not be able to distinguish the causal effect from pleiotropy, and the assumption is violated. Such an extreme situation cannot be identified by any method.

We have adapted the description of the method in the main text and the methods, with the aim to make the description of the assumptions cleaner.

Between lines 127-140 the main text now reads (changed text in red font):

“

The modeling of the pleiotropic variance allows MR-link-2 to be robust to violations of ~~the LD-induced pleiotropy assumption which can bias MR estimates particularly when using only a single genetically associated region. We designed MR-link-2 to work in scenarios that we find~~ replaces the exclusion restriction assumption with three other milder and biologically plausible, ~~however, as with any statistical method, MR-link-2 relies on some modeling assumptions. Thei~~ MR-link-2 assumes that all genetic variants in a locus have non-zero genetic effects on the exposure and their mean is zero (infinitesimal model) ii) all variants (can) have an independent pleiotropic effects on the outcome (InSIDE assumption) and iii) MR-link-2 assumes that the LD matrix is measured without error. We therefore estimate that the likelihood function of MR-link-2 can be sensitive to three parameters: i) when the LD matrix is measured with imprecision (either due to small sample size of the reference panel or population mismatch), ii) when ~~there is a large amount of LD between underlying causal SNPs~~ the pleiotropic effects are correlated and iii) when ~~there is only a small number of SNPs have a non-zero causal SNPs underlying effect on~~ the exposure and the outcome trait.

“

And we have appended the following section to the MR-link-2 likelihood function description to ensure that these assumptions are represented here as well (lines 1068-1077).

“

We estimate that under the following assumptions, the MR-link-2 model provides accurate estimates of causality

1. The relevance assumption (IV-1): The genetic variant G is associated to the exposure X
2. The independence assumption (IV-2): The genetic variant G is independent of any confounder C
3. The infinitesimal genetic architecture assumption: All genetic variants G in a locus have a non-zero genetic effect on the exposure with zero mean

4. Independence of pleiotropy assumption: The pleiotropic effect magnitude $G \rightarrow Y$ are independent from the direct effects of $G \rightarrow X$ (InSIDE)
5. The correlation between genetic variants is measured without error

“

In the meantime, I will also practice some positive and negative controls using the details on the github page.

Response 1.4:

We have now included all the metabolite true positives in the github code repository, under the following link:

https://github.com/adriaan-vd-graaf/mrlink2/tree/main/metabolite_true_positives

Reviewer #1 (Remarks on code availability):

I will have applied the code in time for considering the next iteration of the manuscript.

Response 1.5

We thank the reviewer for their careful consideration of the manuscript and the code.

Reviewer #2 (Remarks to the Author):

Please find my comments to the authors in the attached pdf file "ReviewMRlink2".

Authors note: we copied the text from the PDF and pasted it here.

Review for “MR-link-2: pleiotropy robust cis Mendelian randomization validated in four independent gold-standard datasets of causality”

In this paper, the authors proposed a novel MR method called “MR-link-2”, which is an extension of the existing method “MR-link”. MR-link-2 utilizes summary statistics of correlated SNPs, and is robust to pleiotropy. The authors also compared MR-link-2 to other cis-MR methods via simulations and real data analysis. The three real datasets can be used to quantify the type-I errors of different methods. The proposed method and datasets have certain contributions to the MR field. Please find my detailed comments listed below.

Response 2.1:

We thank the reviewer for their consideration of the manuscript, and their kind words. We hope to have addressed all the comments below.

1). Between line 140 to 145, “the number of causal SNPs underlying both the exposure and the outcome (m_{causal})”. Here, “causal SNPs underlying both the exposure and the outcome” is a bit confusing, do these SNPs have direct (causal) effects only on the exposure, or on both the exposure and outcome? And in the derivation of the method, the number of SNPs is denoted as “ m ”. What is the difference between m_{causal} and m ? Relatedly, in the simulation setup, $m_{causal} \in \{1, 3, 5, 10, 100\}$, however $m = 2068$. Does that mean only m_{causal} SNPs rather than all $m = 2068$ SNPs being used in each simulation? And when $m_{causal} = 1$, is only 1 SNP being used? Some clarifications about these details would be helpful.

Response 2.2:

Thank you for your careful consideration of the simulations. The methods assumes that all variants in the locus are causal for the exposure (i.e. has non-zero multivariable effect). All these SNPs are allowed to have pleiotropic direct effects on the outcome, but their effect sizes are assumed to be independent of those on the exposure. The number of the considered SNPs at the locus is denoted by m .

In the simulations, on the other hand, we break this assumption and out of the m total SNPS select a set of m_{causal} SNPs to have a causal effect on the exposure, and another set of m_{causal} SNPs are selected to have a causal effect on the outcome. These SNPs can be in linkage disequilibrium (LD) with each other. In the simulations, the correlation that occurs

between causal variants is the source of pleiotropy. Still, in the simulations, association summary statistics are generated for all m markers since in realistic settings, we do not know which variants are causal.

We have edited the text in the main text accordingly. The section at line 150 – 159 now reads (changes in red font):

“

Across **2,7003,240** simulation parameter settings, we simulated 1,000 instances of exposure and outcome pairs that are genetically regulated by a single region based on LD derived from the UK10K cohort. We varied 6 parameters: the simulated causal effect (α), the *cis* heritability of the exposure ((h_X^2)) and the extent of pleiotropy of the outcome ((h_Y^2)), the number of causal **SNPs underlying both variants (m_{causal}) that have a genetic effect on the exposure, and the same number of causal (m_{causal}) genetic variant that affect the outcome (~~(m_{causal})~~), their minimum correlation between the causal markers for the exposure and those with direct causal effect on the outcome ($\min(r_{causal})$) and imprecision in the LD reference (parameterized by the reference panel size, n_{ref}) **(Methods)**. **Note that in case of sparse genetic effects, increasing the $\min(r_{causal})$ parameter leads to the violation of the InSIDE assumption.****

”

We further clarify this in the methods, where the new simulation section (line 1086-1112) now reads:

“

We performed extensive simulations to ensure that MR-link-2 provides accurate causal inference, as well as to compare it to other *cis* methods. Our simulations were performed with the goal of mimicking a *cis* region of a molecular -omics study that is potentially causal to a complex trait that is measured in a large cohort. Therefore, the exposure is measured in 10,000 individuals (n_X), while the outcome is measured in 300,000 individuals (n_Y) in a genomic region of 2,068 **SNPs genetic variants (m)** that is derived from a UK10K region on chromosome 10. Our simulations contain six different parameters that we vary: the causal effect ($\alpha \in \{0, 0.505, 0.1, 0.2\}$), the exposure heritability ($(h_X^2 \in \{0.001, 0.01, 0.1\})$), pleiotropy that is represented as outcome heritability ($(h_Y^2 \in \{10^{-20}, 10^{-5}, 10^{-4}, 0.001, 0.01, 0.03\})$), The size of the linkage disequilibrium (LD) reference ($(n_{ref} \in \{500, 5000, \infty\})$), the number of underlying causal SNPs (**$(m_{causal} \in \{1, 3, 5, 10, 100\})$;**) **which represents a subset of m ,** the minimum and maximum LD between causal and pleiotropic SNPs ($(r_{causal}^2 \in \{(0.1, 0.95), (0.01, 0.95), (0.0, 1.0)\}$ for minimum and maximum correlation respectively). In total we have simulated **2,7003,240** different scenarios with 1,000 replications per scenario. Of note, none of the **2,7003,240** parametrizations of our simulations do not violate the specific assumptions MR-link-2's underlying assumptions are violated when $n_{ref} \neq \infty$, $m_{causal} \neq 2,068 = m$ and when $r_{causal}^2 \neq (0.0, 1.0)$.

We simulated summary statistics for the two phenotypes in the following way:

1. **From the total number of markers m , a subset of m_{causal} SNPs are selected from the region for the exposure and the outcome. Selection is random across the region when $r_{causal} = (0, 1)$ and following the procedure of the original MR-link manuscript otherwise. In this procedure, SNPs are selected iteratively until m_{causal} SNPs are selected. **The first** First, a SNP causal to the exposure is drawn randomly from the region, **afterwards** then the next exposure SNP is drawn from all possible SNPs that meet the correlation criteria compared to ~~all other~~ the previously selected SNPs. **The SNPs causal to the outcome will be selected to be within the LD window of at least one exposure causal SNP****

“

It is possible to review the code for the simulations here:

<https://github.com/adriaan-vd-graaf/mrlink2/tree/main/simulations>

(2). Between line 155 and 160: $r^2 > 0.1$ does not seem to be strong LD; and what does “simulating 1 causal SNP for both traits” mean?

Response 2.3.1 regarding the magnitude of correlation:

We thank the reviewer of their question, and apologize for any inconsistency in the manuscript. In our simulations, we select one or more causal SNPs that are within the locus. When simulating 2 causal SNPs with a minimum LD (r^2) of 0.1, the average LD between exposure SNPs across the 100 simulations is much higher, 0.222, which corresponds to an absolute Pearson correlation of 0.47, the actual LD distribution is shown in **(Response 1, Figure 4)**.

Response 1, Figure 4 Linkage disequilibrium between 2 causal snps chosen by our algorithm over 1000 simulations.

When we simulate 100 causal SNPs following the same procedure the median value of the maximum squared pairwise correlation (r^2) of about 0.73 and the whole distribution is shown in **(Response 1 Figure 5)**. Which indicates that there is substantial LD present between causal genetic markers in both cases.

Response 1, Figure 5 Maximum pairwise LD among 100 causal genetic markers across 1000 simulations.

Response 2.3.2, regarding the ‘simulating one causal SNPs for two traits’

We apologize for the typo and any lack of clarity, with ‘simulating one causal SNPs for two traits’, we mean that one direct causal SNP is chosen for an exposure, and one is chosen for the outcome. We have adapted the terminology in lines 173-175 to reflect this (changes in red):

“

However, MR-link-2 is not dependent on the number of causal SNPs that underlie a trait (max T1E rate: 0.05 when simulating 1 causal SNP for ~~both traits~~ **the exposure and 1 causal SNP for the outcome**) (**Supplementary Table 1**)

“

(3). Between line 925 to 940: It seems there are $4 \times 3 \times 6 \times 3 \times 5 \times 3 = 3240$ simulation setups, rather than 2700. And it says that “MR-link-2’s underlying assumptions are violated when $n \neq \infty, \dots$ ”, what are these assumptions explicitly?

Response 2.4:

2.4.1 regarding the number of simulations.

We thank the reviewer for this find, also asked by reviewer 4. The 2,700 number was from an old combination of parameters we tested. In the manuscript that was initially submitted, we performed 3,240 simulations, and these are all shown in Supplementary Table 1. We have adapted the text to reflect the 3,240 simulations. Lines 150-151 in the main text now reads updated text in red:

“

Across ~~2,700~~**2,7003,240** simulation parameter settings, we simulated 1,000 instances of exposure and outcome pairs that are genetically regulated by a single region based on LD derived from the UK10K cohort

”

And the updated method text at lines 1098-1110 now reads:

“

In total we have simulated ~~2,700~~**2,7003,240** different scenarios with 1,000 replications per scenario. Of note, none of the ~~2,700~~**2,7003,240** parametrizations of our simulations do not violate the specific assumptions MR-link-2's underlying assumptions are violated when $n_{ref} \neq \infty$, $m_{causal} \neq 2,068 = m$ and when $r_{causal}^2 \neq (0.0, 1.0)$.

”

2.4.2 regarding the underlying assumptions

We believe that MR-link-2 can make a correct causal estimate under two original IV assumptions, namely the i) relevance assumption, ii) the independence assumption, and three assumptions specific to the MR-link-2 model: iii) Infinitesimal genetic effect assumption, i.e., all genetic variants in a locus have a (small) non-zero genetic effect. iv) The magnitude of pleiotropic effects across all markers is distinct, and v) The LD matrix is generalizable to the populations from which the summary statistics are derived from.

The $n_{ref} \neq \infty$ refers to the precision of the LD matrix estimation, i.e. n_{ref} can be viewed as the sample size of a reference panel from which the LD matrix is estimated and hence n_{ref} is inversely proportional to the LD estimator's variance. We have adapted the text to be more explicit about these assumptions. between lines 127-140 the main text now reads (changed text in red font):

“

The modeling of the pleiotropic variance allows MR-link-2 to be robust to violations of ~~the LD-induced~~ pleiotropy ~~assumption~~ **which can bias MR estimates particularly when** using only a

single genetically associated region. ~~We designed MR-link-2 to work in scenarios that we find~~ replaces the exclusion restriction assumption with three other milder and biologically plausible, ~~however, as with any statistical method, MR-link-2 relies on some modeling~~ assumptions. ~~The~~ MR-link-2 assumes that all genetic variants in a locus have non-zero genetic effects on the exposure and their mean is zero (infinitesimal model) ii) all variants (can) have an independent pleiotropic effects on the outcome (InSIDE assumption) and iii) MR-link-2 assumes that the LD matrix is measured without error. We therefore estimate that the likelihood function of MR-link-2 can be sensitive to three parameters: i) when the LD matrix is measured with imprecision (either due to small sample size of the reference panel or population mismatch), ii) when ~~there is a large amount of LD between underlying causal SNPs~~ the pleiotropic effects are correlated and iii) when ~~there is only~~ a small number of SNPs have a non-zero causal ~~SNPs underlying effect on~~ the exposure and the outcome trait.

“

And we have appended the following section to the MR-link-2 likelihood function description to ensure that these assumptions are represented here as well (lines 1068-1077).

“

We estimate that under the following assumptions, the MR-link-2 model provides accurate estimates of causality

1. The relevance assumption (IV-1): The genetic variant G is associated to the exposure X
 2. The independence assumption (IV-2): The genetic variant G is independent of any confounder C
 3. The infinitesimal genetic architecture assumption: All genetic variants G in a locus have a non-zero genetic effect on the exposure with zero mean
 4. Independence of pleiotropy assumption: The pleiotropic effect magnitude $G \rightarrow Y$ are independent from the direct effects of $G \rightarrow X$ (InSIDE)
- The correlation between genetic variants is measured without error

“

(4). Between line 150 to 160, “up to 0.42 when the LD reference is measured only in 500 individuals”, and “...T1E rate increased to 0.84...”. Type-I errors such as 0.42 and 0.84 are very large, they seem to indicate some problems about the method. Again, it relates to the assumptions of MR-link-2. It is better to provide some more detailed discussions about these highly inflated type-I errors together with some potential issues of the method due to its assumptions.

Response 2.5:

We thank the reviewer for this question. While high type-I error rates are undesirable, these scenarios represent extreme conditions unlikely to occur in real-world datasets: In the

case of the 0.84 type 1 error, it relates to a pleiotropic locus with one underlying causal SNP for the exposure and the outcome, where the locus explains 3% of outcome heritability and 10 % of exposure heritability, and the SNPs are highly correlated to each other. This is an extreme situation when all models simply fail: other methods tested have T1Es that are substantially higher in this scenario (MR-IVW:0.95, MR-IVW-LD: 0.973 and MR-PCA: 0.96). We have explained this in the text and it now reads (lines 175-181 edited parts in red):

“

When violating all these assumptions together, the T1E rate increased to 0.84 when simulating a single causal SNP combined with an extremely large (h^2 of 0.03 (**Supplementary Table 1**), ~~even though this situation is unlikely to occur in human biology, as it is highly unusual to find single variants with such a large effect on complex outcomes.~~). Nonetheless, this is an extreme and unrealistic situation when all other tested *cis* MR methods (MR-IVW T1E: 0.95, MR-IVW-LD T1E: 0.973 and MR-PCA T1E: 0.96) fail and the T1E is still the lowest for MR-link-2

“

In the case of the 0.42 T1E when the LD matrix is measured with a lot of imprecision. In this scenario all LD methods that depend on an LD matrix have a T1E higher than MR-link-2: 0.816 for MR-PCA and 0.782 for MR-IVW LD. MR-IVW has a relatively uninflated T1E with 0.127, as it is not dependent on the LD matrix. We have changed this in the text (lines 166-171):

“

T1E rates generally increased when simulating violations in the MR-link-2 assumptions. MR-link-2 has increased T1E rates when we introduce imprecision in the LD reference (~~up~~. Up to 0.42 when the LD reference is measured only in 500 individuals). This is also seen for other LD dependent methods: MR-PCA T1E: 0.816, MR-IVW LD T1E: 0.782, but as expected, not for MR-IVW, T1E 0.127, which has lower power in exchange (**Supplementary Table 1**)

”

(5). I am not sure whether or not I understand the Figure 3(c) correctly. In Figure 3(c), the 287 reactions across 154 metabolites are treated as ground truth. However, as shown in this figure, some pairs of traits are indirectly associated. For example, for three traits A,B,C, if $A \rightarrow B$ and $B \rightarrow C$, then the marginal relation between A and C is $A \rightarrow C$. In such case, $A \rightarrow C$ should also be a true causal pair, as the proposed MR-link-2 aims to infer pairwise causal relations. And also, there are some loops in Figure 3(c). Are these issue being considered in defining 287 reactions in Figure 3(c)?

Response 2.6.1:

We thank the reviewer for this intriguing question. In previous iterations of the analyses, we actually considered these indirectly causally related traits ($A \rightarrow C$, under the causal graph $A \rightarrow B \rightarrow C$). The reviewer correctly points out that A is indeed causal to C .

In this case, we estimated that there is not enough power to detect causal relationships that are not direct reactions. Detection power is a function of (among others) causal effect magnitude, which need to be multiplied with one another: $\alpha_{A \rightarrow C} = \alpha_{A \rightarrow B} \cdot \alpha_{B \rightarrow C}$. Thus $\alpha_{A \rightarrow C} \leq \alpha_{A \rightarrow B}$ and $\alpha_{A \rightarrow C} \leq \alpha_{B \rightarrow C}$ as any $\alpha \in [-1, 1]$. For example if $\alpha_{A \rightarrow B} = 0.1$ and $\alpha_{B \rightarrow C} = 0.1$, then the causal effect $\alpha_{A \rightarrow C} = 0.01$, for which there is much lower detection power than for direct links. Thus, the detection of these indirect edges is very difficult and hence adding these to the truth set only reduces TPs and hence AUC. In **Response 1 Figure 6**, we are showing the discriminative ability in terms of Area under the receiver operator characteristic curve of taking more a reaction distance ≥ 1 .

Response 1 Figure 6. Area under the receiver operator curve (AUC) at different reaction distances considered as true positives (TP). As the reaction distance increases, the AUC decreases. The true negative reaction distance chosen here is 10 to ensure that true positives and true negatives are not too far apart.

Important to note is that these reactions with reaction distance 1 to 4 are *not* part of the true negative dataset, as the minimum for this is a minimum reaction distance of 5 i.e. the reaction $A \rightarrow F$ is considered a true negative in the graph $A \rightarrow B \rightarrow C \rightarrow D \rightarrow E \rightarrow F$.

We have further specified this in the discussion section, which now reads (line 571-574):

“

We purposefully did not include metabolite pairs that are more than 1 reaction away from each other, as such links would represent an order of magnitude smaller causal effects, which are undetectable given the current sample sizes. Hence such seemingly true links would only be detected by mistake by any method.

”

The reviewer is also correct in stating that there are circular subgraphs that are part of the large graph of figure 3c. We consider a bidirectional causal edge $A \rightarrow B$ and $B \rightarrow A$ as two separate true positives. We added this specification to the Methods section (lines 1310-1311).

“

Of note, sometimes a causal reaction is bidirectional, which translates to both forward and backward causal links being considered as ground truth.

”

(6). In the title, the authors described the four datasets as “gold-standard”. However, the first dataset is simulated, and trait pairs in the other three real datasets are not all well accepted. Some more objective descriptions about the four datasets would be helpful.

Response 2.7:

We used the gold standard datasets nomenclature, as there are only a limited number of molecular datasets that provide a source of true positives in human biology. We believe that by compiling these datasets, we provide an important contribution to the field that investigators can test their observational causal methodology to. For this reason, we feel the nomenclature is warranted, however, as this is not widely approved yet, we changed the ‘gold standard’ nomenclature to ‘independent reference datasets’. Additionally, we reduced the number of datasets mentioned to three by removing the simulations as a reference dataset in the title.

The title is now (lines 1-2):

“

MR-link-2: pleiotropy robust *cis* Mendelian randomization validated in ~~four~~three independent ~~gold-standard~~reference datasets of causality “

Reviewer #2 (Remarks on code availability):

The software provided on GitHub is clear. It also provides a REAME file, instructions, and some examples.

Response 2.8:

We thank the reviewer of their consideration of the code, and appreciate their friendly words.

Reviewer #3 (Remarks to the Author):

The authors present a new version of their previously developed method MR-link, MR-link2. The new version is meant to address single region associations. The authors report around 30% improvement in accuracy compared to selected algorithms, and present results on both simulated and an interesting example on metabolomics data.

While improvements in Mendelian randomization techniques are important for addressing specific challenges, there are some major concerns as follows:

Response 3.1

We thank the reviewer for their careful consideration of the manuscript and their kind words, we hope that the answers below will provide an adequate response to the concerns posed by the reviewer.

1-The proposed improvement of cis MR-link2 as it is presented here is not a significant improvement: Specifying that the current version mainly addresses the single association region is not considered a significant improvement. Also, addressing LD and pleiotropy have already been part of the authors' first version. There needs to be a clear difference between the two versions and the enhancement should be clarified in the abstract and in results.

Response 3.2

We thank the reviewer for their question. One important thing to note is that MR-link (version 1) required individual level data for the outcome, while the new method is applicable to publicly available summary statistics. Since large cohorts with both clinical and omics datasets are scarce, we consider this is substantial advantage. We specified this limitation of the previous version of MR-link in the introduction of the original manuscript (lines 95-97):

“

In contrast to the original MR-link (v1), MR-link-2 does not require individual level data but ~~can~~ it is designed to be ~~used with~~ applied to association summary statistics of ~~trait~~ the exposure and the outcome, allowing for more widespread applications

”

To make it clearer, we added a sentence for this in the description of the MR-link-2 method in the main text. The results text now reads (lines 121-124, changes in red):

“

MR-link-2 is a likelihood function that estimates three parameters based on the exposure and the outcome summary statistics in a region, combined with a reference linkage disequilibrium (LD) matrix (**Figure 1b**) (**Methods**). **MR-link-2 is now more widely applicable as it does not require individual level data, which was a limitation of the original version.**

”

2- Comparisons to previous methods should be more extensive as there are many sophisticated MR methods recently published, for example how does the new version perform in comparison with MR PRESSO (<https://www.nature.com/articles/s41588-018-0099-7>), MR APSS (<https://www.pnas.org/doi/full/10.1073/pnas.2106858119>), and others? Including more algorithms for comparison will be more convincing to researchers and make it more attractive to be used in the future, which is the main aim of the work.

Response 3.3

We thank the reviewer in suggesting these MR methods for comparisons. MR-APSS and MR-PRESSO are powerful methods but require multiple associated regions, which is not the case when performing *cis* MR. Moreover, often eQTLs are only available in *cis* and not genome-wide (see for example the largest eQTL summary statistics from the eQTL-Gen consortium). Notably, this is the reason we developed MR-link-2. MR-PRESSO and MR-APSS cannot make an estimate when there is only one associated region available, as is the case for the metabolite network analysis. As the complex trait true positive combinations have multiple associated loci, we have applied MR-APSS and MR-PRESSO to the complex trait combinations in the following way.

Using the ‘MRAPSS’ and ‘MRPRESSO’ R packages, we clumped ($R^2 < 0.001$, 1000Kb window) the summary statistics at $P < 5 \cdot 10^{-8}$ for MR-PRESSO (genome wide significance), and $P < 5 \cdot 10^{-5}$ for MR-APSS (as suggested by the MR-APSS authors). We applied these methods to the true positive and true negative complex trait combinations also present in our paper.

We find that MR-APSS and MR-PRESSO perform similarly in the true positive combinations, detecting 9 / 9 combinations at a nominal P value threshold (0.05), however both have wider confidence intervals than more tailored *cis*-MR methods (**Response 1 Figure 6**).

Response 1 figure 6 True positive complex—>complex trait combinations. All true positive trait combinations are found at nominal significance.

In the true negative combinations, similarly to MR-PCA and MR-link-2, they only detect HDL-Cholesterol -> Stroke, and HDL-Cholesterol->coronary artery disease (**Response 1 Figure 7**).

Response 1 figure 7 True negative complex onto complex trait combinations. Fully colored hexagons are significant at a nominal significance level.

We have adapted the main text to reflect these results in the complex trait analysis as follows (lines 412-423):

“

~~After applying *cis* MR methods~~ Since complex traits do not rely on MR performed in a single region, we additionally explored the performance of state-of-the-art non-*cis* MR methods, such as the MR-PRESSO and the MR-APSS methods, to five trait combinations that are unlikely to be causal, e.g., outcomes that precede the risk factor in time, such as adult LDL-C levels impacting childhood onset asthma (COA) (**Methods**). Only one false positive link is identified at nominal significance (by MR-PCA, LDL-CIVW and MR-IVW LD, diastolic blood pressure on COA, $P=0.02023$ and $P=0.019$ respectively) (**Supplementary Table 11**) (**Methods**). ~~Perhaps as expected, all~~ All methods falsely identify the ‘non-causal’ relationship between HDL-C and coronary artery disease (CAD) as well as between HDL-C and stroke, which is notoriously difficult to accurately estimate through univariable MR methods⁹ (**Figure 5a**). In the positive control analysis, all MR methods including MR-APSS and MR-PRESSO identify a causal relationship at nominal significance (**Supplementary Table 11**).

”

We also applied MR-APSS and MR-PRESSO to the cell type specific gene expression directional dataset, where MR-APSS exhibits good discriminative ability compared to all other methods, only being outperformed by MR-link-2 in identifying the correct effect direction between cell types and gene expression datasets (**Response 1 Figure 8**).

Response 1 Figure 8 Discriminative ability of all the methods tested in this study to identify the correct causal effect direction between cell type abundance and gene expression levels.

We have included these results into figure 5f in the main text.

3-The write up of the manuscript needs much improvement. For example, one cannot get clear messages from the results section as it is currently a mix of results, discussion and methods. Results should be more clear, focused and ideas organized.

Response 3.4

We thank the reviewer for their suggestion on improving the clarity of the message of the manuscript. Due to space limitations, we sometimes combined the results with discussion, and methodology. We have changed the results section of the main text to contain less methodology and discussion, while remaining aware of the intricacies of the methodology.

We have changed the following sections to reduce discussion of results if they were not essential to the continuation of the manuscript. We provide a few examples of improved clarity below, but have changed the manuscript in other sections as well.

We have changed the following sections (lines 176-178) (changes in red.)

“

When violating all these assumptions together, the T1E rate increased to 0.84 when simulating a single causal SNP combined with an extremely large h^2 0.03 (**Supplementary Table 1**), ~~even though this situation is unlikely to occur in human biology, as it is highly unusual to find single variants with such a large effect on complex outcomes.-)~~

”

We changed (lines 207-209):

“

Unfortunately not many causal relationships are established for molecular exposures that could be used to reliably benchmark MR methods. Here, we compile reactions between metabolites as true positive molecular causal relationships. ~~We believe this is a promising avenue since some of them have been known for more than 85 years and have been extensively experimentally validated.~~ We use these causal relationships as a ground truth, to understand when a particular MR methods fail and to subsequently compare these MR methods to each other.

”

We changed (lines 238-245) :

“

Indeed, when considering Bonferroni significant MR estimates (~~218,163238,097~~ testable exposure, outcome and associated region combinations, $P < 2.31 \cdot 10^m$), all methods identify more positive effects than negative effects (range: ~~5958%~~80%) (**Figure 4e-h**), with the highest percentage (80%) for MR-link-2 (**Figure 4e**), considerably higher than the second-best performing method, PCA-MR (~~63%~~) (**Figure 4h**). ~~Together these analyses indicate that the significant MR-link-2 estimates represent metabolism better than the significant estimates of other cis MR methods (Figure 4e-h62%) (Figure 4h) (Supplementary Table 6) (Supplementary Text).~~

”

We changed (lines 261-268):

“

If we compare the discriminative performance of the MR methods with coloc methods, we find that in aggregate, coloc methods have lower AUC than any MR method used here (**Figure 4k-p**) (**Table 1**). ~~Of note, generally the discriminative performance of coloc SuSIE is better in our simulations, however, to ensure that methods use roughly the same amount of data, we fall back to the original coloc method when coloc SuSIE does not identify multiple causal variants).~~ This results in similar discriminative performance across comparisons (Pearson r: ~~0.969997~~). The AUC of MR-IVW and MR-IVW LD are also very correlated (Pearson r: ~~0.992~~), ~~as the LD corrected method produces identical results when there is only a single IV detected. Two methods-998).~~

”

We removed the following two paragraphs (lines 284-298), as we felt it did not make the main message of the paper more clear:

“

~~The discriminative performance of MR methods also allows us to assess the characteristics of these pathway reference datasets. Considered individually, the MetaCyc pathway has the highest median AUC (0.684, MR-link-2), followed by KEGG (0.648, MR-link-2) and WikiPathways (0.639, MR-link-2) (Figure 4d). When combining these pathway references, the intersection of all pathways has the highest median AUC (0.705, MR-PCA). However, the intersection is also the most unstable with the highest standard deviation across minimum reaction distances (maximum for coloc: 0.061) (Supplementary Table 7). Indeed, the variability of the AUC estimates generally increases with the minimum reaction distance used for the null edge definition, which is where most rank changes between AUCs of methods were found (Figure 4k-p). Indicating that as the datasets of false causal links reduce in size, the stability of the discriminative ability estimate also decreases.~~

~~Interestingly, in the KEGG and MetaCyc pathway references, the discriminative ability of MR-methods initially increases as the shortest path length to define null edges increases, coming to a plateau (Figure 4k, l), suggesting that MR methods may have some power to detect metabolites that are linked through multiple reactions. Generally, MR-link-2 has increased discriminative ability over the other methods tested in this study.~~

”

We changed (lines 462-467):

“

~~Moreover, when meta-analysis of analyzing all loci for all complex trait combinations analyzed in this study, we find that MR-link-2 has substantially lower heterogeneity in terms of Cochran's Q statistic (median for MR-link-2: 586582, lowest competing: 13591021) (Methods) (Figure 5d) (Supplementary Table 11). The low heterogeneity statistic of meta-analysis could be attributed to MR-link-2's low false positive rates, which have more realistic standard error estimates, possibly due to accounting for pleiotropic effects.~~

”

These are changes where we removed discussion of the results when they were not relevant for the continuation of the manuscript narrative. These and other changes are present in the edited manuscript, that we hope will increase the clarity of the manuscript.

Other minor concerns are attached as a pdf.

Response 3.5:

We thank the reviewer for suggesting other minor concerns which we are responding to in combination with the response to reviewer 4 below.

Reviewer #4 (Remarks to the Author):

Response 4.1

We thank the reviewer for their careful consideration of the manuscript, and would like to commend the reviewer for their diligent code review. We have pasted the text of the PDF questions below, and are responding to the questions in this document.

1- In python code (GitHub: mr_link_2_standalone.py), Why there is two versions for loglik_reference (v0 (without using exponential) and v2 (with exponential)

2 - In the Python code (GitHub: mr_link_2_standalone.py), for the function loglik_reference_v2, why was the exponential used in the calculation of DXX, DYY, and DXY, and then the logarithm taken? This step was not included in the equation in the paper. As appeared below.

Response 4.1

Thank you for asking this important question, we have used the two functions for the following reason: Sometimes, the v_0 function would emit a floating point error warning for otherwise reasonable values of α , σ_y and σ_x and λ because they would be rounded to zero, which will produce illegal values of infinity when used as a denominator in division.

Therefore, we have used a common numerical trick to avoid division by zero errors that are due to numerical precision.

Consider a value x , and then we take the exponent and the logarithm of x , this is equal to $x = \exp(\log(x))$. If x is defined as a multiplication, $x = a*b$, this allows us to rewrite as $x =$

$\exp(\log(a) + \log(b))$, as $\log(a*b) = \log(a) + \log(b)$. Similarly, if x represents a division: $x = a/b$, this can be represented by $\log(a/b) = \log(a) - \log(b)$

In the helpful image Reviewer #4 has attached in the document, they highlight occurrences of this trick of red boxes, we will discuss all of these boxes from top to bottom.

Top red box:

In the case of the definition of the Dxx parameter:

$$Dxx = 1. / (((a ** 2 * n_y + n_x) * lam + tX) - a ** 2 * n_y ** 2 * (lam ** 2) / (n_y * lam + tY))$$

Is converted into:

$$Dxx = 1. / (np.exp(np.log(a ** 2 * n_y + n_x) + np.log(lam)) + tX - np.exp(np.log(a ** 2 * n_y ** 2 * (lam ** 2)) - np.log(n_y * lam + tY)))$$

We use this trick twice in the assignment of Dxx, first to ensure that the $(a ** 2 * n_y + n_x) * lam$ term doesn't become too large or too small when large 'a' values are provided, or very small 'lam' values are provided to the function: $(np.exp(np.log(a ** 2 * n_y + n_x) + np.log(lam)))$. Secondly to ensure that in the $a ** 2 * n_y ** 2 * (lam ** 2) / (n_y * lam + tY)$ that the denominator doesn't become too small leading to infinite values: $np.exp(np.log(a ** 2 * n_y ** 2 * (lam ** 2)) - np.log(n_y * lam + tY))$

So even though in principle the V0 function is correct, we edited the function into the v2 function to ensure that less divide by zero warnings are emitted. To ensure that both functions output the same log likelihood, we've tested the two versions together in the github repository to ensure that there is no deviation between the two.

Please find these tests under the following link:

https://github.com/adriaan-vd-graaf/mrlink2/blob/main/tests/unit_tests_mr_link_2.py

3- In python code (github: mr_link_2_standalone.py), regarding log-likelihood function, the function written in the code is different compared to the paper in page 38 line

895

Response 4.2

We thank the reviewer for pointing out these two inconsistencies. We very much appreciate the scrutiny that the reviewer has applied to our manuscript's math and code.

Considering the topmost ellipse in the likelihood function:

Thank you for this great find! Unfortunately a typo in the description of the likelihood function slipped into the manuscript. The derivation above states that c_y^{**2} / lam needs to be normalized by $n_y/2$. We have updated the likelihood function accordingly in the main text.

The updated likelihood function now reads (lines 1045-1053):

“

$$\begin{aligned}
 L(\hat{\beta}_X, \hat{\beta}_Y | \alpha, \sigma_X^2, \sigma_Y^2) = & -m \cdot \log(2\pi) \\
 & - \frac{1}{2} \cdot \sum_{i=1}^m \log \left((\alpha^2 \cdot n_Y + n_X) \cdot \lambda_i + \sigma_X^{-2} - \frac{\alpha^2 \cdot n_Y^2 \cdot \lambda_i^2}{n_Y \cdot \lambda_i + \sigma_Y^{-2}} \right) \\
 & - \frac{1}{2} \cdot \sum_{i=1}^m \log(n_Y \cdot \lambda_i + \sigma_Y^{-2}) \\
 & + \frac{1}{2} \sum_{i=1}^m ((\hat{\delta}^{(X)})_i)^2 \cdot D_{(i,i)}^{(X,X)} + \sum_{i=1}^m \hat{\delta}_i^{(X)} \cdot \hat{\delta}_i^{(Y)} \cdot D_{(i,i)}^{(X,Y)} \\
 & + \frac{1}{2} \sum_{i=1}^m ((\hat{\delta}^{(Y)})_i)^2 \cdot D_{(i,i)}^{(Y,Y)} \\
 & - \frac{n_X}{2} \sum_{i=1}^m ((\hat{\delta}_i^{(X)})^2 / \lambda_i) - \frac{n_Y}{2} \sum_{i=1}^m ((\hat{\delta}_i^{(Y)})^2 / \lambda_i) + \frac{m}{2} \cdot (\log(n_X) + \log(n_Y)) \\
 & - \sum_{i=1}^m \log(\lambda_i) - m \cdot (\log(\sigma_X) + \log(\sigma_Y))
 \end{aligned}$$

”

Luckily this mistake only happened once and was not carried over to the rest of the derivation, hence all final results and code are correct and remain the same.

Regarding the bottom ellipse:

The difference between the text and the likelihood function is that the manuscript uses the parameter σ_Y and σ_X , whereas, we represent the tX and tY parameter as m/σ^2 in the

python likelihood function, which is more convenient as it does not result in too small floating point parameters that need to be optimized.

We've updated the documentation of the function to make sure that this is fairly represented in the function signature. The function signature now reads:

```
"""
The MR-link2 log likelihood function. This function calculates  $-1 * \text{likelihood}$  of
three parameters:
alpha, sigma_x and sigma_y.
Designed to be used in optimization algorithms like those in scipy.minimize

:param th:
    List or numpy array of floats with the parameters to optimize first is alpha,
    second the
    1 / exposure heritability and third the 1/ outcome heritability.
:param lam:
    np.ndarray of selected eigenvalues of the cX and cY parameters.
:param c_x:
    The dot product of the selected eigenvectors and summary statistics vector of the
    exposure
:param c_y:
    The dot product of the selected eigenvectors and summary statistics vector of the
    outcome
:param n_x:
    The number of individuals in the exposure dataset
:param n_y:
    The number of individuals in the outcome dataset
:return:
    a single float that contains the likelihood of the parameters theta.
"""
```

4- It is not clear how the normalization was done in the paper and in the code (GitHub: `mr_link_2_standalone.py`) in function “`mr_link2_on_region`”

Response 4.4:

Thank you for this important question, normalization happens quite early in the pipeline. The relevant code can be found here:

https://github.com/adriaan-vd-graaf/mrlink2/blob/d19b276e9b5559d185cc1e425b65ce59e58c33a8/mr_link_2_standalone.py#L1672 where the exposure is normalized

and here:

https://github.com/adriaan-vd-graaf/mrlink2/blob/d19b276e9b5559d185cc1e425b65ce59e58c33a8/mr_link_2_standalone.py#L1847

where the outcome is normalized.

5- In page 26, the direction between the SNP and confounders needs to be reversed.

Response 4.5

Thank you for pointing this out. Unfortunately, it is a common misconception that the arrow between the IV and the confounder should go from IV to the confounder and it is discussed extensively in Carter and Anderson (7): (<https://doi.org/10.1093/ije/dyae050>).

If the arrow points the other way, it would constitute a form of (correlated) pleiotropy, while the arrow in the other direction represents population stratification/dynastic effects, etc.

6- In page 40, line 975, what is the reference of those equations

Response 4.6

We thank the reviewer for noticing this inconsistency in the equation. We have now attached a derivation of these equations attached to the supplementary notes.

7- In page 39, line 930. In Simulations paragraph, the alpha effect needs to be changed to 0.05.

Response 4.7

Thank you for the finding of this typo, We have updated the manuscript accordingly. The updated line now reads (lines 1092-1093, updated word in red).

“

Our simulations contain six different parameters that we vary: the causal effect ($\alpha \in \{0, 0.05, 0.1, 0.2\}$),

”

8- in page 39, line 935. It was mentioned that there were 2,700 different scenarios but it was 3240 in the excel file.

Response 4.8

We thank the reviewer for their careful consideration of our simulation section, in an older version of a manuscript draft, we performed 2,700 simulations, however, we increased this to 3,240. The new manuscript now reflects this, while the underlying data remains the same. Lines 150-151 in the main text now reads (updated text in red):

“

Across ~~2,700~~^{3,240} simulation parameter settings, we simulated 1,000 instances of exposure and outcome pairs that are genetically regulated by a single region based on LD derived from the UK10K cohort

”

And the updated method text at lines 1098-1110 now reads:

“

In total we have simulated ~~2,700~~^{3,240} different scenarios with 1,000 replications per scenario. Of note, none of the ~~2,700~~^{3,240} parametrizations of our simulations do not violate the specific assumptions MR-link-2's underlying assumptions are violated when $n_{ref} \neq \infty$, $m_{causal} \neq 2,068 = m$ and when $r_{causal}^2 \neq (0.0, 1.0)$.

”

9- It will be good if there is a link for the simulation code.

Response 4.9:

We have now added the simulation code to the github, you can find this here:

<https://github.com/adriaan-vd-graaf/mrlink2/tree/main/simulations>

10- Page 48, What is the reference of next equations

Response 4.10:

These equations are a normalization modeling assumption of the MR-link-2 likelihood function. As the genetic region can contain widely differing amounts of genetic variants, we modeled σ_Y and σ_X independent of the number of SNPs in a region. We have updated the methods text to ensure that this is now fairly represented. The updated text now reads (lines 1023-1028) :

“

Here, G represents a genotype matrix, with normalized genotypes to zero-mean and unit variance across samples. SNP effects are modelled as random, $\gamma^{(X)} \sim N(0, \sigma_X^2)$ and $\gamma^{(Y)} \sim N(0, \sigma_Y^2)$, where σ_X^2 is a **per variant heritability estimate** related to the *cis* heritability of X , such that $\sigma_X^2 = h_X^2/m$ and σ_Y^2 is the **per variant** direct (vertically pleiotropic) *cis* heritability of Y , such that $\sigma_Y^2 = h_Y^2/m$.

“

11- In page 37, line 845, reference 23 was putted on both techniques “MR-PCA” and “MR-IVW LD”, could you please check if it is correct.

Response 4.11

We thank the reviewer for their careful checking of all the references. In this paper, multiple methods are proposed, MR-IVW LD and MR-PCA. The code for the MR-IVW LD method can be found on page 11 of the original paper (page 724 of the journal), preceded by “IVW estimate (accounting for correlation):” and the code for the MR-PCA method can also be found on page 724 of the original paper preceded by “IVW estimate (accounting for correlation) using principal components:”

12- The authors should compare the performance of the proposed techniques with

others techniques which deals with horizontal pleiotropy such as this technique

“Mendelian randomization for causal inference accounting for pleiotropy and sample

structure using genome-wide summary statistics”

Link: <https://www.pnas.org/doi/full/10.1073/pnas.2106858119>

Response 4.12

Thank you for this important suggestion also requested by Reviewer 3, we refer to response 3.3, where we implemented MR-APSS and MR-PRESSO on the complex trait analyses, and on the cell type specific analysis.

Reviewer #4 (Remarks on code availability):

The code review is represented in my attached report

Response to reviewer 4 code review,

We have tried to answer all the questions in the responses above, we hope that they will be considered favorably.

References:

1. Morrison J, Knoblauch N, Marcus JH, Stephens M, He X. Mendelian randomization accounting for correlated and uncorrelated pleiotropic effects using genome-wide summary statistics. *Nature Genetics*. 2020 Jul;52(7):740–7.
2. Sun BB, Chiou J, Traylor M, Benner C, Hsu YH, Richardson TG, et al. Plasma proteomic associations with genetics and health in the UK Biobank. *Nature*. 2023 Oct;622(7982):329–38.
3. Said S, Pazoki R, Karhunen V, Võsa U, Ligthart S, Bodinier B, et al. Genetic analysis of over half a million people characterises C-reactive protein loci. *Nat Commun*. 2022 Apr 22;13(1):2198.
4. Initiative TC 19 HG, Ganna A. Mapping the human genetic architecture of COVID-19 by worldwide meta-analysis. *medRxiv*. 2021 May 8;2021.03.10.21252820.
5. Budu-Aggrey A, Kilanowski A, Sobczyk MK, Shringarpure SS, Mitchell R, Reis K, et al. European and multi-ancestry genome-wide association meta-analysis of atopic dermatitis highlights importance of systemic immune regulation. *Nat Commun*. 2023 Oct 4;14(1):6172.
6. van der Graaf A, Claringbould A, Rimbart A, Westra HJ, Li Y, Wijmenga C, et al. Mendelian randomization while jointly modeling cis genetics identifies causal relationships between gene expression and lipids. *Nat Commun*. 2020 Oct 1;11(1):4930.
7. Carter AR, Anderson EL. Correct illustration of assumptions in Mendelian randomization. *International Journal of Epidemiology*. 2024 Apr 1;53(2):dyae050.

Please find below our comments to the second round of comments on our manuscript titled: “MR-link-2: pleiotropy robust cis Mendelian randomization validated in three independent reference datasets of causality”.

We would like to thank the reviewers for sharing their comments, and helping us meaningfully improve the paper. We have highlighted the reviewer comments with blue marking, and commented on all the comments on a point by point basis.

Thank you again and best wishes,
Zoltán Kutalik

REVIEWERS' COMMENTS

Reviewer #1 (Remarks to the Author)

I thank the authors for addressing my comments.

IL6R inhibitors for COVID-19 are a clinically validated positive control and the failure of the MR-link 2 method to identify this effect needs to be acknowledged as a limitation for analyses that are of direct translational relevance.

If practitioners relied on MR-link 2, they would not have identified the critical genetic evidence that led to tocilizumab being prioritized for treating covid-19 during the pandemic.

Response 1.1

We thank the reviewer for highlighting the translational importance of IL6R inhibitors in the context of COVID-19. Indeed, MR-link-2 did not detect this clinically validated positive control and that this limitation should be made explicit for readers who seek direct translational insights.

We would like to point out that the IL-6 region is not genetically associated to COVID-19 at genome wide significance in the data (see **Response 2 Figure 1**). Furthermore, the top associations do not colocalize visually (**Response 2 Figure 1**) nor statistically (coloc PP4 $1.0 \cdot 10^{-32}$). We have therefore revised the passage to address two main points more clearly: (1) the robust clinical evidence supporting tocilizumab and related IL6R inhibitors for COVID-19, and (2) the fact that our lack of detection may reflect other causal mechanisms compared to the the IL6R locus for CRP → COVID-19, as the genetic evidence is only conclusive at the lower significance threshold in the other MR methods.

Response 2 Figure 1

Locus plot of the IL6 gene region + / 250 Kb for Severe COVID-19 vs healthy population (European individuals, **Methods**) and C reactive protein. Including gene tracks (genome build 37, ensembl).

The passage used to read:

“

This undetected positive control (C-reactive protein → COVID-19 instrumented from IL6R) has little evidence of causal variant colocalization (coloc PP4: $1.0 \cdot 10^{-32}$) in the locus, suggesting that the positive control may not be as convincing as prior studies indicate. None of the methods detect the negative controls at this lenient multiple testing threshold (Supplementary Table 14).

”

We have adapted the passage to now read:

“

This undetected positive control (C-reactive protein → COVID-19 instrumented from IL6R) shows little evidence of causal variant colocalization (coloc PP4: 1.0×10^{-32}) in this locus. This finding contrasts with the clinically validated efficacy of tocilizumab in severe COVID-19, which targets the IL-6 pathway <ref>. One possible explanation is that the CRP genetic signal at this locus does not fully capture the relevant immunopathological mechanisms of IL-6 in COVID-19. Only at the lowest thresholds of significance (0.05/6) was this combination significant in MR-IVW LD and MR-PCA.

”

Reviewer #2 (Remarks to the Author)

Thanks the authors for their efforts to address the comments I raised. I have no further comment about the revised manuscript, thanks.

Response 2.1

We thank the reviewer for their careful consideration of the work, and their comments that meaningfully improved the manuscript.

Reviewer #3 (Remarks to the Author)

Thanks for the detailed response with more experimental evidence and for the clarifications. There are very few minor comments and one main concern regarding the code consistency with the paper equations (a follow up on point 4.2) as pointed out in the attached document. Please clarify that as it may cause confusion between the paper and the code.

Response 3.1

We thank Reviewer #3 for their comments, the answering of which has definitely improved the work. We take the opportunity to comment on the attached document in our response to Reviewer #4.

(Remarks on code availability)

Reviewer 4 has revised the code corresponding to the equations and we included that in the previous revision and this round of revisions are included in the attached document. There is an inconsistency between the paper and the part of the code that we mentioned in the last comment in the attached review (point 4.2), where we highlight the inconsistency. It is important that the authors revise this part and clarify it in the paper and code.

Response 3.2

We thank Reviewer #3 for their comments. We take the opportunity to comment on the attached document in our response to Reviewer #4.

Reviewer #4 (Remarks to the Author)

(Remarks on code availability)
include in the pdf

Response 4.1

We thank the reviewer for their important contributions to the work, particularly identifying the typo in the main equation of the original manuscript, and the careful scrutiny of the code, which has helped improve the manuscript, and improved the trust and readability of the manuscript.

We copy over the text from the PDF and respond to the comments here.

Regrading response 3.3, I observed that the performance of the techniques used are the same except that the MR-APSS and MR-PRESSO have large confidence interval compared to other techniques. It is important to discuss in the paper whether those results are specific to this particular dataset.

Response 4.2

We thank the reviewer for their keen assessment that MR-APSS and MR-PRESSO have larger confidence intervals compared to the *cis* methods tested in this work, which is also stated in the first reviewer response letter. Based on the reviewer comment, we have now adapted the main text to reflect this.

The main text now reads:

“

In the positive control analysis, all MR methods including MR-APSS and MR-PRESSO identify a causal relationship at nominal significance, albeit with larger confidence intervals compared to the *cis* methods (Supplementary Table 11).

“

This observation is not limited to the complex traits, which are actually derived from multiple independent studies, as the cell type to complex analysis also exhibits this (**Response 2 Figure 2**, Supplementary Table 14 of the original paper).

Response 2, Figure 2

Distribution of standard errors per MR method tested of the positive controls in the cell type to gene expression analysis.

Regarding Response 4.1, adding two versions of the model, one in the paper and the other in the code is confusing, so we suggest that it might be better to present the version that uses the exponentiation and justify it in the main paper.

Response 4.2

We thank the reviewer for their valuable suggestion. We believe that including the currently listed ‘simplified’ model in the paper is valuable as it provides the reader with a more clean intuition on how parameters change the variables of the paper. We do agree that tracing back the function in python with the equation may provide some confusion, and therefore, we’ve taken the liberty to add an ‘implementation details’ section to the derivation pdf, where these implementation details are further specified. Please find a copy of the implementation details in the supplementary note. We’re copying over the relevant text section here. A copy of the code is also shown on the next page.

Implementation details

We have implemented the MR-link-2 likelihood function. To ensure numerical stability, we have made two purely numerical transformations in the likelihood function. First we $t_X = \sigma_X^{-2}$ and $t_Y = \sigma_Y^{-2}$. This will reduce numerical instability when either σ_X or σ_Y are very close to zero.

The likelihood function of MR-link-2 that is optimized is then:

$$\begin{aligned}
 l(\hat{\beta}^{(x)}, \hat{\beta}^{(y)}) &= l(\hat{\delta}^{(x)}, \hat{\delta}^{(y)}) \\
 &= -m \cdot \log(2\pi) - \frac{1}{2} \cdot \sum_{i=1}^m \log \left((\alpha^2 \cdot n_y + n_x) \cdot \lambda_i + t_X - \frac{\alpha^2 \cdot n_y^2 \cdot \lambda_i^2}{n_y \cdot \lambda_i + t_Y} \right) \\
 &\quad - \frac{1}{2} \cdot \sum_{i=1}^m \log (n_y \cdot \lambda_i + \sigma_y^{-2}) \\
 &\quad + \frac{1}{2} \left(\sum_{i=1}^m (\hat{\delta}_i^{(x)})^2 \cdot D_{i,i}^{(x,x)} + 2 \cdot \sum_{i=1}^m \hat{\delta}_i^{(x)} \cdot \hat{\delta}_i^{(y)} \cdot D_{i,i}^{(x,y)} + \sum_{i=1}^m \hat{\delta}_i^{(y)} \cdot D_{i,i}^{(y,y)} \right) \\
 &\quad - \frac{n_x}{2} \cdot \left(\sum_{i=1}^m (\hat{\delta}_i^{(x)})^2 / \lambda_i \right) - \frac{n_y}{2} \cdot \left(\sum_{i=1}^m (\hat{\delta}_i^{(y)})^2 / \lambda_i \right) \\
 &\quad + \frac{m}{2} \cdot (\log(n_x) + \log(n_y)) - \sum_{i=1}^m \log(\lambda_i) + \frac{m}{2} \cdot (\log(t_X) + \log(t_Y))
 \end{aligned}$$

To further reduce numerical instability of the likelihood function, we make use of the exponent logarithm trick to reduce numerical instability: Consider a value x , and then we take the exponent and the logarithm of x , this is equal to $x = \exp(\log(x))$. If x is defined as a multiplication, $x = a * b$, this allows us to rewrite as $x = \exp(\log(a) + \log(b))$, as $\log(a * b) = \log(a) + \log(b)$. Similarly, if x represents a division: $x = a / b$, this can be represented by $\log(a / b) = \log(a) - \log(b)$

We also perform simplifications when $\alpha = 0$. A copy of the python function is found on the next page

Regarding Response 4.2: The equation in the paper is inconsistent with the equation in the Python code. t_x was replaced with both σ_x^{-2} and σ_x similarly to t_y (as highlighted below). Please clarify that.

We thank the reviewer for their continued scrutiny in the equation and the model. We enumerate the 4 red annotation boxes from top to bottom and left to right.

The python variables t_X and t_Y are defined as σ_X^{-2} and σ_Y^{-2} in the likelihood function respectively, this makes it easy to define the first 2 boxes: **Box 1**: σ_X^{-2} is converted directly into t_X and **Box 2**: σ_Y^{-2} is converted directly into t_Y .

Then, using the following equality ($\log(n^{-2}) = -2 \log(n)$), we can rewrite the $-m \cdot \log(\sigma_X)$ in **Box 3**: $-m \cdot \log(\sigma_X) = +\frac{m}{2} \cdot \log(\sigma_X^{-2}) = +\frac{m}{2} \cdot \log(t_X)$. We do the same for **Box 4**: $-m \cdot \log(\sigma_Y) = +\frac{m}{2} \cdot \log(\sigma_Y^{-2}) = +\frac{m}{2} \cdot \log(t_Y)$. Then, in the function, the $+\frac{m}{2}$ factor is combined: $+\frac{m}{2} \cdot \log(t_X) + \frac{m}{2} \cdot \log(t_Y) = +\frac{m}{2} \cdot (\log(t_X) + \log(t_Y))$, which can be found in the code: `m/2 * np.log(tX) + np.log(tY)`.

Review for “MR-link-2: pleiotropy robust cis Mendelian randomization validated in four independent gold-standard datasets of causality”

In this paper, the authors proposed a novel MR method called “MR-link-2”, which is an extension of the existing method “MR-link”. MR-link-2 utilizes summary statistics of correlated SNPs, and is robust to pleiotropy. The authors also compared MR-link-2 to other cis-MR methods via simulations and real data analysis. The three real datasets can be used to quantify the type-I errors of different methods. The proposed method and datasets have certain contributions to the MR field. Please find my detailed comments listed below.

- 1i). Between line 140 to 145, “the number of causal SNPs underlying both the exposure and the outcome (m_{causal})”. Here, “causal SNPs underlying both the exposure and the outcome” is a bit confusing, do these SNPs have direct (causal) effects only on the exposure, or on both the exposure and outcome? And in the derivation of the method, the number of SNPs is denoted as “ m ”. What is the difference between m_{causal} and m ? Relatedly, in the simulation setup, $m_{causal} \in \{1, 3, 5, 10, 100\}$, however $m = 2068$. Does that mean only m_{causal} SNPs rather than all $m = 2068$ SNPs being used in each simulation? And when $m_{causal} = 1$, is only 1 SNP being used? Some clarifications about these details would be helpful.
- (2). Between line 155 and 160: $r^2 > 0.1$ does not seem to be strong LD; and what does “simulating 1 causal SNP for both traits” mean?
- (3). Between line 925 to 940: It seems there are $4 \times 3 \times 6 \times 3 \times 5 \times 3 = 3240$ simulation setups, rather than 2700. And it says that “MR-link-2’s underlying assumptions are violated when $n_{ref} \neq \infty, \dots$ ”, what are these assumptions explicitly?
- (4). Between line 150 to 160, “up to 0.42 when the LD reference is measured only in 500 individuals”, and “...T1E rate increased to 0.84...”. Type-I errors such as 0.42 and 0.84 are very large, they seem to indicate some problems about the method. Again, it relates to the assumptions of MR-link-2. It is better to provide some more detailed discussions about these highly inflated type-I errors together with some potential issues of the method due to its assumptions.
- (5). I am not sure whether or not I understand the Figure 3(c) correctly. In Figure 3(c), the 287 reactions across 154 metabolites are treated as ground truth. However, as shown in this figure, some pairs of traits are indirectly associated. For example, for three traits A, B, C , if $A \rightarrow B$ and $B \rightarrow C$, then the marginal relation between A and C is $A \rightarrow C$. In such case, $A \rightarrow C$ should also be a true causal pair, as the proposed MR-link-2 aims to infer **pairwise causal relations**. And also, there are some loops in Figure 3(c). Are these issue being considered in defining 287 reactions in Figure 3(c)?

- (6). In the title, the authors described the four datasets as “gold-standard”. However, the first dataset is simulated, and trait pairs in the other three real datasets are not all well accepted. Some more objective descriptions about the four datasets would be helpful.

- 1- In python code (GitHub: mr_link_2_standalone.py), Why there is two versions for loglik_reference (v0 (without using exponential) and v2 (with exponential)
- 2- In the Python code (GitHub: mr_link_2_standalone.py), for the function loglik_reference_v2, why was the exponential used in the calculation of DXX, DYY, and DXY, and then the logarithm taken? This step was not included in the equation in the paper. As appeared below.

Equation in paper

$$D^{(X,X)}, D^{(X,Y)} \text{ and } D^{(Y,Y)} \text{ are diagonal matrices with diagonal elements defined as } D_{(i,i)}^{(X,X)} = \left(\left((\alpha^2 \cdot n_y + n_x) \cdot \lambda_i + \sigma_X^{-2} \right) - \frac{\alpha^2 \cdot n_Y^2 \cdot \lambda_i^2}{n_Y \cdot \lambda_i + \sigma_Y^{-2}} \right)^{-1}, \quad D_{(i,i)}^{(X,Y)} = -D_{(i,i)}^{(X,X)} \cdot \frac{\alpha \cdot n_Y \cdot \lambda_i}{n_Y \cdot \lambda_i + \sigma_Y^{-2}} \text{ and } D_{(i,i)}^{(Y,Y)} = \frac{1}{n_Y \cdot \lambda_i + \sigma_Y^{-2}} + D_{(i,i)}^{(X,X)} \cdot \frac{\alpha^2 \cdot n_Y^2 \cdot \lambda_i^2}{(n_Y \cdot \lambda_i + \sigma_Y^{-2})^2}.$$

Equation in python

```

if a != 0.0:
    Dxx = 1. / (np.exp(np.log(a ** 2 * n_y + n_x) + np.log(lam)) + tX -
               np.exp(np.log(a ** 2 * n_y ** 2 * (lam ** 2)) - np.log(n_y * lam + tY)))
    Dxy = -Dxx * a * np.exp(np.log(n_y * lam)) - np.log(n_y * lam + tY)
    Dyy = Dyy + np.exp(np.log(Dxx * (a ** 2 * n_y ** 2 * lam ** 2)) - (2 * np.log(n_y * lam + tY)))
    asq_ny_sq_lam_sq_div_ny_lam_ty = np.exp(np.log(a ** 2 * n_y ** 2 * (lam ** 2)) - np.log(n_y * la
else:
    Dxx = 1. / (np.exp(np.log(n_x) + np.log(lam)) + tX)
    Dxy = -Dxx * a * np.exp(np.log((n_y * lam)) - np.log(n_y * lam + tY))
    Dyy = Dyy
    asq_ny_sq_lam_sq_div_ny_lam_ty = 0.0 * lam

```

- 3- In python code (github: mr_link_2_standalone.py), regarding log-likelihood function, the function written in the code is different compared to the paper in page 38 line 895

Equation in python

```

loglik = -m * np.log(2 * np.pi) + \
    -(1 / 2) * sum(np.log((a ** 2 * n_y + n_x) * lam + tX - asq_ny_sq_lam_sq_div_ny_lam_ty)) + \
    -(1 / 2) * sum(np.log(n_y * lam + tY)) + \
    +(1 / 2) * (sum(dX ** 2 * Dxx) + 2 * sum(dX * dY * Dxy) + sum(dY ** 2 * Dyy)) + \
    -(n_x / 2) * sum((c_x ** 2) / lam) + \
    -(n_y / 2) * sum((c_y ** 2) / lam) + \
    +(m / 2) * (np.log(n_x) + np.log(n_y)) - sum(np.log(lam)) + (m / 2) * (np.log(tX) + np.log(tY))

```

Equation in paper

$$\begin{aligned}
 L(\hat{\beta}_X, \hat{\beta}_Y | \alpha, \sigma_X^2, \sigma_Y^2) &= -m \cdot \log(2\pi) \\
 &- \frac{1}{2} \cdot \sum_{i=1}^m \log \left((\alpha^2 \cdot n_Y + n_X) \cdot \lambda_i + \sigma_X^{-2} - \frac{\alpha^2 \cdot n_Y^2 \cdot \lambda_i^2}{n_Y \cdot \lambda_i + \sigma_Y^{-2}} \right) \\
 &- \frac{1}{2} \cdot \sum_{i=1}^m \log(n_Y \cdot \lambda_i + \sigma_Y^{-2}) \\
 &+ \frac{1}{2} \sum_{i=1}^m ((\hat{\delta}^{(X)})_i)^2 \cdot D_{(i,i)}^{(X,X)} + \sum_{i=1}^m \hat{\delta}_i^{(X)} \cdot \hat{\delta}_i^{(Y)} \cdot D_{(i,i)}^{(X,Y)} \\
 &+ \frac{1}{2} \sum_{i=1}^m ((\hat{\delta}^{(Y)})_i)^2 \cdot D_{(i,i)}^{(Y,Y)} \\
 &- \frac{n_X}{2} \left(\sum_{i=1}^m ((\hat{\delta}_i^{(X)})^2 / \lambda_i) + \sum_{i=1}^m ((\hat{\delta}_i^{(Y)})^2 / \lambda_i) \right) \\
 &+ \frac{m}{2} \cdot (\log(n_X) + \log(n_Y)) \\
 &- \sum_{i=1}^m \log(\lambda_i) - m \cdot (\log(\sigma_X) + \log(\sigma_Y))
 \end{aligned}$$

- 4- It is not clear how the normalization was done in the paper and in the code (GitHub: `mr_link_2_standalone.py`) in function “`mr_link2_on_region`”.

```

## these should have been normalized. In my case they are.
exp_betas = np.asarray([snp_to_beta_dict[x][0] for x in ordered_snps], dtype=float)
exp_pvals = np.asarray([exp_dict[x][3] for x in ordered_snps], dtype=float)

out_betas = np.asarray([snp_to_beta_dict[x][1] for x in ordered_snps], dtype=float)
out_pvals = np.asarray([out_dict[x][3] for x in ordered_snps], dtype=float)

```

- 5- In page 26, the direction between the SNP and confounders needs to be reversed.

6- In page 40, line 975, what is the reference of those equations

$$\beta_{standardized} = \frac{z}{\sqrt{n + z^2}}$$
$$se(\beta_{standardized}) = \frac{1}{\sqrt{n + z^2}}$$

7- In page 39, line 930. In Simulations paragraph, the alpha effect needs to be changed to 0.05.

the causal effect ($\alpha \in \{0, 0.5, 0.1, 0.2\}$),

8- in page 39, line 935. It was mentioned that there were 2,700 different scenarios but it was 3240 in the excel file.

total we have simulated 2,700 different scenarios with 1,000 replications per scenario.

9- It will be good if there is a link for the simulation code.

10- Page 48, What is the reference of next equations

$$\text{thus } h_x^2 = \sigma_x^2 \cdot \tilde{m} \text{ and } h_y^2 = \sigma_y^2 \cdot m.$$

11- In page 37, line 845, reference 23 was putted on both techniques “MR-PCA” and “MR-IVW LD”, could you please check if it is correct.

12- The authors should compare the performance of the proposed techniques with others techniques which deals with horizontal pleiotropy such as this technique

“Mendelian randomization for causal inference accounting for pleiotropy and sample structure using genome-wide summary statistics”

Link: <https://www.pnas.org/doi/full/10.1073/pnas.2106858119>